# Investigation of Autonomous Multi-UAV Systems for Target Detection in Distributed Environment: Current Developments and Open Challenges

Wilfried Yves Hamilton Adoni [1,2,*,†] , Sandra Lorenz [2,†] , Junaidh Shaik Fareedh [2,†] , Richard Gloaguen [2,†] and Michael Bussmann [1,†]

1   Helmholtz-Zentrum Dresden-Rossendorf, Center for Advanced Systems Understanding, Untermarkt 20, 02826 Görlitz, Germany
2   Helmholtz-Zentrum Dresden-Rossendorf, Helmholtz Institute Freiberg for Resource Technology, Chemnitzer Str. 40, 09599 Freiberg, Germany; s.lorenz@hzdr.de (S.L.); j.shaik-fareedh@hzdr.de (J.S.F.)
*   Correspondence: w.adoni@hzdr.de; Tel.: +49-351-2604487
†   These authors contributed equally to this work.

**Abstract:** Uncrewed aerial vehicles (UAVs), also known as drones, are ubiquitous and their use cases extend today from governmental applications to civil applications such as the agricultural, medical, and transport sectors, etc. In accordance with the requirements in terms of demand, it is possible to carry out various missions involving several types of UAVs as well as various onboard sensors. According to the complexity of the mission, some configurations are required both in terms of hardware and software. This task becomes even more complex when the system is composed of autonomous UAVs that collaborate with each other without the assistance of an operator. Several factors must be considered, such as the complexity of the mission, the types of UAVs, the communication architecture, the routing protocol, the coordination of tasks, and many other factors related to the environment. Unfortunately, although there are many research works that address the use cases of multi-UAV systems, there is a gap in the literature regarding the difficulties involved with the implementation of these systems from scratch. This review article seeks to examine and understand the communication issues related to the implementation from scratch of autonomous multi-UAV systems for collaborative decisions. The manuscript will also provide a formal definition of the ecosystem of a multi-UAV system, as well as a comparative study of UAV types and related works that highlight the use cases of multi-UAV systems. In addition to the mathematical modeling of the collaborative target detection problem in distributed environments, this article establishes a comparative study of communication architectures and routing protocols in a UAV network. After reading this review paper, readers will benefit from the multicriteria decision-making roadmaps to choose the right architectures and routing protocols adapted for specific missions. The open challenges and future directions described in this manuscript can be used to understand the current limitations and how to overcome them to effectively exploit autonomous swarms in future trends.

**Keywords:** UAV; RPAS; UAS; uncrewed aerial vehicles; drones; multi-UAV systems; autonomous swarm; autonomous aerial vehicles; communication architectures; FANET; routing protocols; collaborative missions; distributed environment; distributed path planning

## 1. Introduction

Uncrewed aerial vehicles (UAVs) or drones [1] belong to the large family of connected objects. After being utilized extensively in military operations [2,3], drones have remained out of reach for civilians [4] due to their high price. With recent innovations in microcontrollers and sensors, drone prices have been reduced to become affordable. Currently, they are widely used for commercial purposes such as surveys, photography, and cinematography. UAVs, in general, are outfitted with on-board sensors that allow them to

collect geospatial information about their environment and are remotely controlled from a ground control station [5]. From this station, the operator can plan and supervise the evolution of the mission. Their use in the civil and industrial sector [4] has allowed us to optimize industrial processes or to carry out missions in hostile environments that are partially or entirely inaccessible to humans [6–9]. For example, in the agricultural field, farmers are now facing various problems that impact the quality of their crops. Drones are an efficient and cheap means to collect information on ecosystems and their variations due to, e.g., climate change, soil erosion, water availability, and meteorological extreme events. They are, for example, also used in spraying plantations, which saves time and optimizes yields [10]. Goodrich Payton et al. [11] have demonstrated the efficiency of drones in precision agriculture. From the collected data, they reconstructed a 3D cartographic representation of the plots to better analyze the density of vegetation and soil heterogeneities.

Drones provide a broad variety of purposes in the health sector, including delivering medical supplies to remote or hard-to-reach areas, e.g., transporting blood samples and lab results [12]. In the field of transportation, drones can be used for package delivery [13–15], traffic monitoring [16], and infrastructure inspections [17]. Drones have also been used to map volcanoes' terrain and to detect volcanic activity. Thiele et al. [5] used drones equipped with thermal cameras, gas sensors, and other instruments to measure temperature, gas concentrations, and other indicators of volcanic activity. This information can be used to predict eruptions, to elaborate rescue operations, for photogrammetry and infrastructure monitoring, or even for delivery services. UAVs can also be used to study the geology of a volcano and its surrounding area, providing valuable information for volcano research. The use of UAVs in this field can greatly improve the efficiency, accuracy, and safety of operations, as well as decrease costs by reducing the need for human intervention in dangerous areas [5]. However, it implies several challenges related to communication services, such the range, security system, and communication architecture [18].

The communication architecture of drones relies on a Flying Ad-Hoc Network (FANET) [19], without the requirement for a fixed infrastructure. FANET is a decentralized ad-hoc network that enables communication between the ground station and flying vehicles, such as drones and aircraft [20]. As shown in Figure 1, FANET inherits from both Vehicular Ad-Hoc Network (VANET) [21] and Mobile Ad-Hoc Network (MANET) [22] networks. It is a subclass of MANET that is an extension to highly mobile devices such as smartphones and laptops. These three types of ad-hoc networks share the ability to form and maintain a network connection dynamically [23]. Combined, they are an effective tool in creating a wide-area network that links UAVs, vehicles, and communication devices. They can also be used for applications such as collision avoidance [24], traffic jam prevention, and intelligent transportation systems [25]. In this manuscript, we will focus on the FANET protocols as well as the different paradigms of data exchange between UAVs, especially for multi-UAV systems.

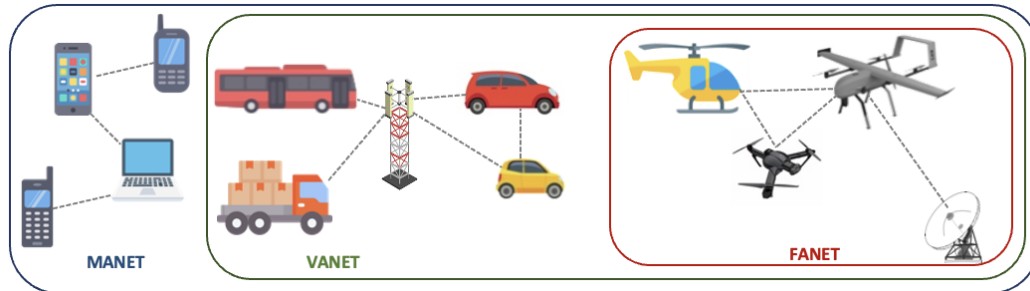

**Figure 1.** Different types of wireless Ad-Hoc Networks: FANET [19] vs. VANET [21] and MANET [22].

As the need for UAVs grows, there is an increasing requirement for systems that can coordinate several UAVs in a dispersed environment [1,26]. This is where multi-UAV systems are relevant. Multi-UAV systems, commonly called swarms, are composed of multiple

UAVs that are coordinated to collaborate in order to accomplish a shared purpose [27]. These systems have the potential to coordinate mission tasks across multiple UAVs in a parallel manner. They have several possible applications, including rescue missions [7,28], surveillance and reconnaissance [6,26,29,30], environmental monitoring [17], and even payload carrying [13–15]. One of the primary benefits of swarms is their capacity to operate cohesively, allowing them to cover large regions quickly and make choices collectively, making them more robust to failure than individual UAVs [31]. Swarms operate according to the design of a multi-UAV system architecture that is controlled by a central system [6] or through decentralized algorithms [1,32,33].

To work in a coordinated manner, a multi-UAV system requires the following key components. (1) The UAVs themselves: these are the drones that make up the swarm. (2) A ground control station: this is the central point of control for the drone swarm. It oversees the transmission of commands to the drones, as well as the retrieval of data from them. It can be a single computer or a cluster of computing nodes. (3) A communication system: it consists of devices and antennas that enable communication between the drones and the ground station via a common protocol. The communication system can use a variety of technologies, including MANET [22], FANET [19], and VANET [21]. (4) A navigation system: this is the system that enables the UAVs to fly and locate themselves in the environment. It includes sensors such as GPS and Inertial Measurement Units (IMUs). (5) A control system: it allows the ground station to control the drones and to coordinate their actions. It consists of software for mission planning, decision making, and swarm behavior. (6) Finally, a data processing system: this is the system that allows the ground station to process the data in real time and feed the data back to the control system. It can include tools for image processing, data analysis, and machine learning. All these components work together to allow the swarm to function as a cohesive unit, with the ground station providing overall command and control, while the UAVs collaborate to attain a common aim.

Depending upon the operations' nature and mission requirements, the architecture of a multi-UAV system might be centralized [6] or decentralized [32,33]. Because of its complexity and the very dynamic environment of operation, there are several challenges related to using multi-UAV for highly mobile networks operating in large-area missions:

- **Communication**: maintaining communication between the drones in a swarm can be challenging, especially in highly mobile and wide areas. The drones must be able to maintain communication even when they are moving at high speeds or when they are far apart from each other.
- **Coordination**: coordinating the actions of the UAVs is complex, especially in elaborate missions. To reach a shared purpose, the drones must be able to successfully collaborate.
- **Autonomy**: the drones must be able to operate autonomously without human intervention. This requires collaborative actions for decision making, navigation, and swarm behavior. Theses challenges include spatial awareness, maintaining a distance from each other, and communicating potential threats to other drones, such as heavy wind gusts, rain, and obstructions.
- **Scalability**: multi-UAV systems must be able to support a large number of drones, they must be reliable, and they must be able to scale up or down depending on the requirements of the mission.
- **Reliability**: the system must be reliable, even if one or more UAVs fail. This requires robust algorithms for fault detection, diagnosis, and recovery.
- **Security**: a multi-UAV system must be encrypted, guarding against hacking, jamming, and other types of interference to the drones and their communications [18,34,35].
- **Interference**: in highly dynamic surroundings, drones may encounter interference from other wireless devices, which can affect communication or navigation [18].
- **Energy consumption**: to ensure that the drones can operate for long periods of time, energy management is crucial. Drones must be able to manage their power consumption and plan their routes to optimize battery life [35–42].

- **Interoperability**: the system must guarantee the exchange of information between different types of drones regardless of their communication protocols. Unfortunately, there is still no common protocol for drones to communicate with each other.

One key challenge in the design and implementation of multi-UAV systems is ensuring that the UAVs work together effectively in a distributed environment. This requires the development of robust communication and coordination architectures, as well as the integration of sensors and other equipment to allow the UAVs to sense and adapt to their environment. One of the main problems with swarms is the coordination of multiple UAVs carrying out a task in a distributed area. This problem can be addressed by using distributed algorithms. Algorithms such as consensus algorithms [43], leader–follower algorithms [44], and distributed optimization algorithms [45] can be used to coordinate multiple drones in a distributed area. These algorithms allow the UAVs to communicate and coordinate with each other so that they can complete the task in an efficient manner. Furthermore, these algorithms can also be used to ensure safety and prevent collisions between UAVs (see Section 10.1). Overall, these challenges require sophisticated algorithms and technologies, and the development of new solutions to these challenges is an active area of research in the field of drone swarm technology.

In this research, we explore the problem of target detection in a distributed environment. We focus on a review of communication architectures and routing protocols, as well as the integration of routing protocols to enable effective coordination among the UAVs. In the second part, we show how to elaborate a centralized and decentralized communication architecture to achieve a collaborative mission. With this work, we aim to contribute to the progress of multi-UAV systems and their potential use in a wide range of applications.

**Innovations**   By leveraging the latest advancements in the current state of the art, this work fills the gaps in the integration from scratch of autonomous multi-UAV systems for collaborative actions. More specifically, the main added value of this work lies in the following points:

- The field of autonomous swarms is still in its infancy and, as a result, there is a lack of research that consistently provides a formalism of the constraints related to the collaborative actions of multi-UAV systems. In this context, our work has added significant value by filling the gap concerning a well-defined mathematical formulation of collaborative actions of multi-UAV systems in a large distributed environment.
- The study of an autonomous and cooperative drone swarm infrastructure is an incredibly complex field, and there are still very few scientific studies that fully explore the intricacies of how it operates. Due to this lack, we conduct an extensive investigation of multi-UAV infrastructures, which results in a scientific work providing valuable knowledge to answer the question of how to choose the right communication architecture as well as the most suitable routing protocol. This will save a lot of time and avoid erroneous conceptual studies.
- Despite the importance of understanding the collaborative paradigm of autonomous multi-UAV systems for target detection in distributed environments, there is a lack of clear visibility of the challenges and future directions in developing effective collaborative swarm. As added value, we have highlighted the most important challenges and clarified the open perspectives for research in this area.

**Objectives**  The goal of this work is to motivate the use of multi-UAV systems by a well-informed study of the constraints related to their implementation. We offer useful suggestions on how to choose the type of UAV, the communication architecture, as well as the type of protocol. These suggestions are based on a multi-criteria survey that considers the complexity of the mission, the environment, and the capabilities of the UAVs used. This work can serve as a background for all types of projects involving the use of multi-UAV systems. In this paper, we formalize the problem of collaborative target detection in large-scale environments. We conduct a comparative analysis of existing communication

architectures and routing protocols for the efficient deployment of a swarm, especially for complex missions. The following are the key contributions.

1. Types of UAVs: We review existing drones. The UAVs are classified into four categories according to their shape and size.
2. Problem formulation: We start by identifying the key features that allow us to recognize a multi-UAV ecosystem. We emphasize the complexities and major challenges associated with the collaborative target detection problem. Then, we propose a mathematical formulation of the problems of k-partition of the environment and path planning to reach the targets. The proposed models are based on multi-objective functions that satisfy the constraints of the workload balancing of the swarm and the optimization of the paths.
3. Architectures and protocols: We give cutting-edge knowledge on multi-UAV communication architectures as well as the existing routing protocols. Then, we evaluate the performance and draw a comparison for each communication architecture and routing protocol listed in this paper.
4. Suggestions: Based on the comparisons of the existing architectures and routing protocols, we give some suggestions for choosing the right architectures and routing protocols that are tailored to a well-defined mission.
5. Open challenges: We discuss the ongoing issues and potential future directions, such as collaborative decision making, communication infrastructure, spatial organization, and regulation for multi-UAV deployment.

**Organization** The rest of this article is structured as follows. Section 2 provides useful information on multi-UAV ecosystems. Subsequently, in Section 3, we perform a comparative study of the types of autonomous uncrewed aerial vehicles. Based on this study, we highlight the advantages and drawbacks of different types of UAVs. In Section 4, we present formal definitions and mathematical explanations about the problem of collaborative target identification. Furthermore, in Section 5, we provide the state of the art of some applications of multi-UAV systems in the civilian domain. Meanwhile, in Sections 6 and 7, we discuss the existing communication architectures and provide a roadmap that helps to choose the most adapted. Similarly, Sections 8 and 9 provide a comparison of routing protocols, followed by a roadmap of how to choose the right protocols. Section 10 discusses the open difficulties and potential directions. Finally, in Section 11, we conclude this work.

## 2. Background

We provide here a detailed description and useful definitions and terminology for the ecosystem of a multi-UAV system.

### Uncrewed Aerial Vehicle (UAV)

An uncrewed aerial vehicle or Remotely Piloted Aircraft (RPA), also referred to as a drone, is an aircraft that does not require a human pilot onboard. It flies autonomously or remotely through an onboard computer, remote control, or a combination of the two. It is a type of Uncrewed Aerial System (UAS), also known as a Remotely Piloted Aircraft System (RPAS) [8], which is defined as an aircraft system that operates without direct human intervention in the aircraft [46]. Most of the time, the interaction with the aircraft is made possible through a ground control station, a remote controller, a data transmission system, and dedicated hardware and software support (see Figure 2).

### Autonomous UAV

Autonomous UAVs are a specific case of aerial vehicles that can fly to a given location without the assistance of a remote human. This type of vehicle is able to perceive, interact, and act with respect to its environment in an autonomous way for decision making on a mission [7].

### Remote Controller (RC)

A remote controller is a device that can be hand-held by a pilot to communicate with a drone via a live stream of data signals.

### Ground Control Station (GCS)

A ground control station is the command post that offers the necessary elements to the human for the remote control of UAVs. A GCS is designed for flight planning and monitoring, as well as the real-time processing of UAV–environment interactions. A GCS is equipped with a Human–Machine Interface that conducts the planning of the missions, display and control consoles, a real-time video stream, an on-board computer, telemetry instrumentation, a data communication platform, and links with the UAVs via antennas [30].

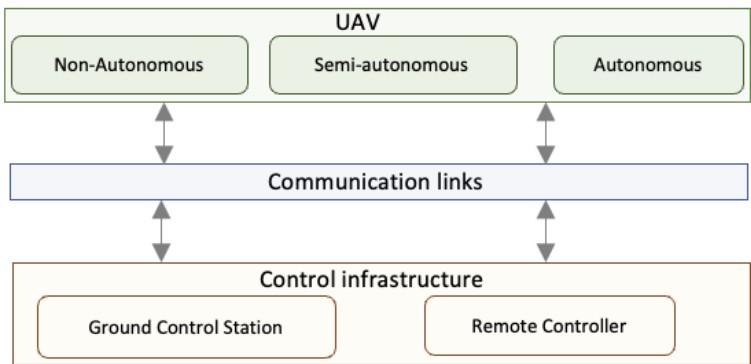

**Figure 2.** Overview of RPAS components.

### Multi-UAV (M-UAV)

Multi-UAV is a group or swarm of UAVs that operate simultaneously to achieve a mission. In the architecture of M-UAV, we distinguish two types of interaction: cooperation and coordination. Coordination [43–45] involves sharing the operational environment with other drones safely. It therefore performs an important role in large space exploration missions involving numerous UAVs. Meanwhile, in the logic of cooperation [47,48], drones collaborate on different tasks to achieve a common goal.

### Environment

An environment represents the exploration domain of the UAV. The domain could be either aerial, terrestrial, or marine (oceanic or aquatic). It can be static or dynamic depending on the natural factors that influence its complexity, such as climatic conditions, weather, vegetation, as well as generic elements.

### Perception

Perception is the input information that the UAV receives through embedded sensors from its environment at any moment in order to make decisions about actions to take. The sensors commonly used for UAV perception are proximity sensors, cameras for object recognition, and laser sensors for range detection.

### Target

A target is a physical entity or an object located in an exploration space. It is considered static if it does not present any form of mobility in its environment or dynamic in the case of spatio-temporal changes.

### Mission

A mission is a task entrusted to a UAV. Drones are increasingly employed in a range of operations, including the following: mapping, exploration, rescue, investigation, reconnaissance, traffic monitoring [16,48], weather monitoring, earth observation, firefighting [5],

agriculture [11], photography, and videography. A mission is said to be collaborative [43–45,47–49] when multiple UAVs work together to accomplish the mission.

**Waypoints**

Waypoints represent a set of intermediate positions through which a UAV must pass to reach a well-known final position. Obviously, a deviation tolerance threshold must be considered.

Figure 3 depicts an interactive view of the ecosystem of a multi-UAV system. It is composed of a ground environment in which there are two categories of targets, i.e., mobile (2 persons and 1 dog) and fixed (trees), and an aerial environment in which there are non-autonomous and autonomous UAVs. The non-autonomous UAVs are guided by an operator from the ground, while the autonomous UAVs carry out their mission without human assistance. Each drone has the capacity for the perception of the environment. It can collaborate with other UAVs (air-to-air). It can also collect data and communicate them to the GCS from the ground (air-to-ground). The mission planning as well as the command and control operations are centralized in the GCS, which stores the data received from the UAVs.

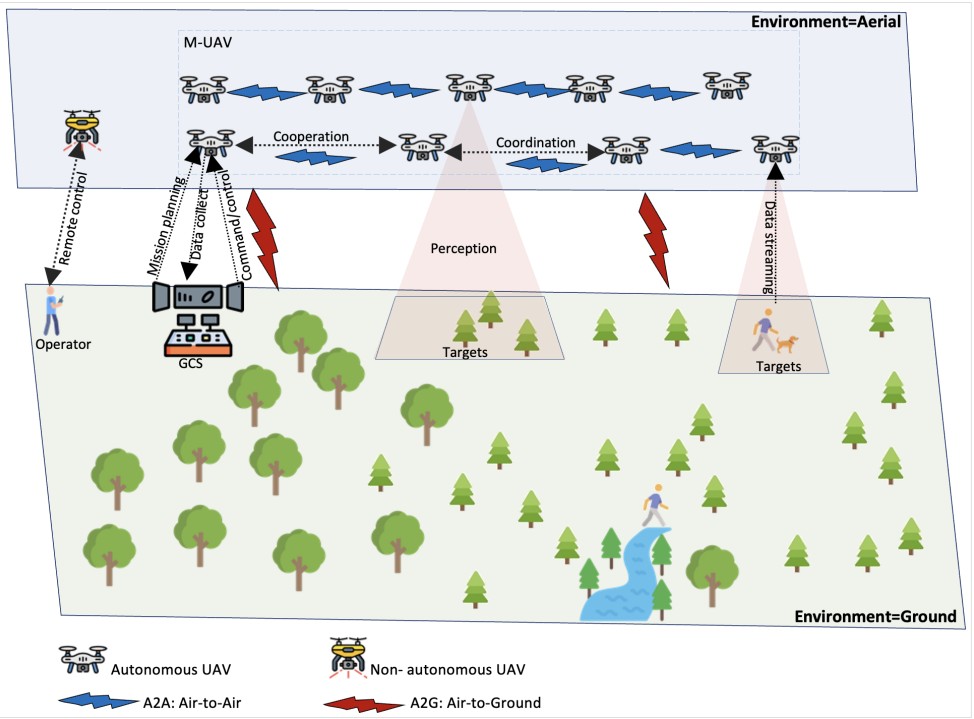

**Figure 3.** Overview of a multi-UAV ecosystem.

## 3. Classification of UAVs

The growing interest in UAVs in recent years has led to the strong emergence of various types of aircraft with varying configurations and components in terms of shape and size. UAVs are divided into four types: single-rotor, multi-rotor, fixed-wing, and hybrid [33], as shown in Figure 4.

1.  **Single-rotor** [50] (or helicopter): this category of UAVs takes off and lands vertically. Generally, they use a main rotor for attitude control (roll, pitch, and yaw) and a tail rotor to control the direction. Its main advantage is the ability to carry heavy payloads over a longer flight time. However, the complexity of their mechanical systems, as well as the large size and high cost of the rotors constitute a danger for the uncrewed versions. It is more unstable in bad weather.
2.  **Multi-rotor** [51] (or multicopter): this is a UAV having more than two rotors. This UAV category is further subdivided into five sub-categories, which are birotor, trirotor,

quadrotor, hexarotor, and eight-rotor (octocopter) [13,15]. As with single-rotor UAVs, multi-rotor UAVs also ensure a vertical takeoff and landing. They are fast and agile in flight, allowing them to perform complex maneuvers and flights in confined spaces. However, the short flight time is the main weakness of these types of aircraft.

3. **Fixed-wing** [52]: the navigation principle of this category of UAVs is based on a simple structure of a fixed rigid wing. The classification of these drones is not only based on the type of wing, but also on the body and the power system (Li-ion, Li-Po batteries, or gas-powered). They are subdivided into four subcategories: normal, swept back, swept forward, and delta. Furthermore, they can carry heavier payloads than multi-rotors [53]. The disadvantage of these UAVs is that they have limited agility in flight, which does not allow them to perform complex maneuvers and fly over confined spaces, as well as the necessity for a runway for takeoff and landing.

4. **Hybrid** [54]: this last category is still under development. It is an improved version that takes advantage of both multi-rotor and fixed-wing UAVs. They offer good agility and velocity on long-distance flights. They can carry large payloads and do not require a runway. The main disadvantages are the high price, the complicated mechanics, and the lower performance in terms of flight stability and the restrictive speed ranges.

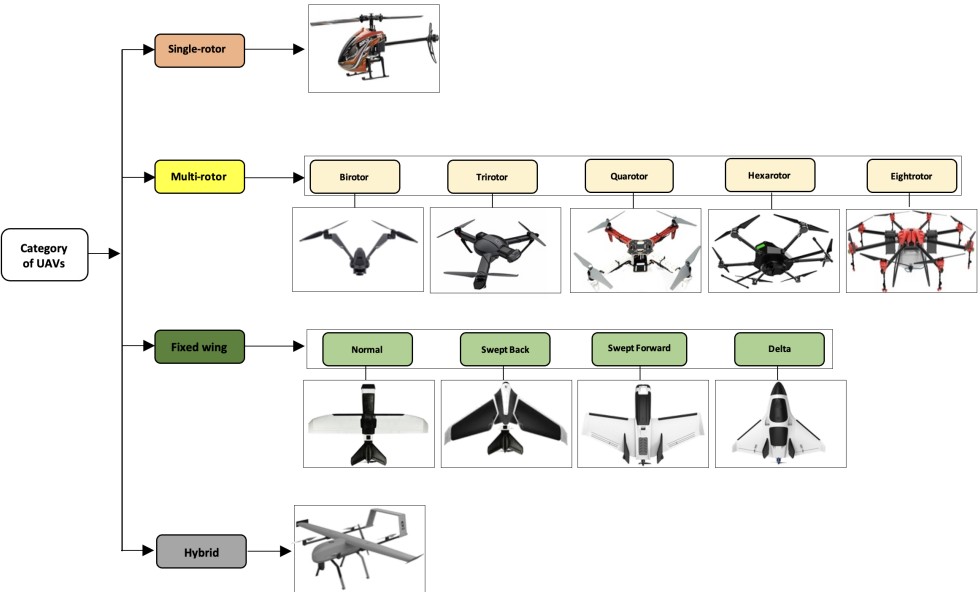

**Figure 4.** Different categories of UAVs.

　　Table 1 presents a brief comparative study of the advantages and disadvantages of each category of UAV. This comparison focuses on payload capacity, endurance, agility, speed, and area of coverage.

　　Generally, the classification is made according to the weight, size, range, and endurance [32]. We will focus on the classification according to the size, which will be useful for this discussion. As shown in Table 2, four classes of UAVs, organized as very small, small, medium, and large, are usually defined.

　　Very small UAVs refer to micro- or nano-UAVs. These UAVs resemble insects or birds with wings, and their dimensions generally vary between 5 and 50 cm. The components are extremely small and lightweight. They can reach a top speed of >10 km/h and are limited to a maximum altitude of ca. 120 m by law. Typical examples of very small UAVs include the German TobyRich SmartPlane Pro (https://www.drohnen.de/tag/tobyrich/, accessed on 15 March 2023) with a wing length of 30 cm; it is ultra-stable and equipped with a VGA camera of $640 \times 480$ pixels. Moreover, the DJI Mavic 3 (https://store.dji.com/de/product/dji-mavic-3, accessed on 15 March 2023), configured with dimensions of $28.3 \times 10.7$ cm$^2$, is

another small model of UAV; it is equipped with a 4:3 CMOS camera and can fly for 46 min with a maximum transmission range of ca. 15 km.

**Table 1.** Comparative study of UAVs: advantages vs. disadvantages.

| Categories | Strengths | Weak Points |
| --- | --- | --- |
| Single-rotor | Heavy payload<br>Flight time<br>Hovering flight<br>VTOL | Mechanical system<br>Rotor size |
| Multi-rotor | Hovering flight<br>VTOL<br>Velocity<br>Agile maneuverability<br>Confined-space flying | Flight time |
| Fixed-wing | Flight time<br>High Velocity<br>Large-area coverage | Complex aerofoil<br>Limited payload<br>No hovering flight<br>Limited maneuverability |
| Hybrid | Flight time<br>Hovering flight<br>VTOL<br>Agile maneuverability<br>Velocity<br>Large-area coverage | Very expensive<br>Unstable transition mechanism from horizontal to vertical flight |

Small UAVs, often known as mini-UAVs, refer to UAVs whose dimensions exceed at least 50 cm and not more than 2 m. The design of these aircraft is essentially based on the fixed-wing model, and the majority are launched by propelling them into the air by the operator. They can carry a maximum payload of 9 kg and fly at a maximum speed of ca. 150 km/h at an altitude usually not exceeding 400 m. The German Wingcopter 198 (https://wingcopter.com/, accessed on 15 March 2023) is an electric VTOL drone used for delivery services. It is designed with a configuration of $198 \times 154$ cm$^2$ with a flight time of 90 min. It is hybrid and can fly in both multicopter and fixed-wing modes. The second one is Astro (https://freeflysystems.com/alta-x, accessed on 15 March 2023), a quadrotor designed with a configuration of $141 \times 51$ cm$^2$ with a flight time of 37 min. It incorporates a gimbaled a7R IV mapping camera and the data transfer is based on the MAVLink communication protocol [55]. The last one is Scorpion (https://www.quantum-systems.com/, accessed on 15 March 2023), a VTOL that has a wingspan of 1.37 m$^2$ and a flight endurance of 35 min. It provides data transfer up to a range of 25 km and it supports a maximum weight of >7 kg.

A UAV is considered "medium" if it is too heavy to be handled by one person but smaller than an aeroplane, since they can only carry a payload of 200 kg. They are usually in the category of fixed-wing UAVs and typically have a wingspan whose length ranges from 5 to 10 m. They can fly at a maximum velocity of 463 km/h without exceeding an altitude of almost 1 km. There are numerous examples of UAVs in this scope of size, such as the Trinity F90+ (https://www.quantum-systems.com/, accessed on 15 March 2023), Vector (https://www.quantum-systems.com/, accessed on 15 March 2023), Alta X (https://freeflysystems.com/alta-x, accessed on 15 March 2023), and, recently, the Yangda YD6-1600S (https://www.yangdaonline.com/, accessed on 15 March 2023). The German intelligent VTOL Trinity F90+ and Vector have average fixed wings of 2.8 m and, respectively, a flight time of 90 min and 120 min. The Alta X is a VTOL quadrotor with a 50 min flight time and it carries a maximum payload of 15 kg for 8 min. It has a peak speed of more than >95 km/h and uses the MAVLink [55] protocol for data transmission. The hexacopter Yangda YD6-1600S is designed with dimensions of $1.6 \times 2.35$ m$^2$. It can reach a cruise speed of >72 km/h and can also carry a maximum payload of 5 kg for a flight time not exceeding 45 min.

The last class of UAVs is mainly used for military purposes. The large UAVs have a wide range and endurance. Furthermore, their designs are usually based on a fixed-wing structure, which allows them to carry heavy payloads over long distances while reaching a maximum altitude of 5.5 km. Some examples of these UAVs are the Czech Primoco UAV One 150 (https://uav-stol.com/primoco-uav-one-150/, accessed on 15 March 2023) and the Chinese Feng Ru 3-100 (https://ev.buaa.edu.cn/info/1133/3165.htm, accessed on 15 March 2023). Primoco is designed to fly for 15 h. It can carry a payload of 50 kg and has a radio range of 200 km. Feng Ru is designed for couriers; it has a wingspan of 19.6 m and can fly for 5 days.

Currently, there is no standard classification of UAVs. In the case of this study, we have focused on size. However, other important parameters can be considered, such as autonomy, speed, range, altitude, and payload capacity.

**Table 2.** UAV classification according to the size (https://www.e-education.psu.edu/geog892/node/5, accessed on 15 March 2023).

| Size | Dimensions $(m^2)$ | Payload (kg) | Velocity (km/h) | Altitude (km) | Example (Accessed on 15 March 2023) |
|---|---|---|---|---|---|
| Very small | 0.3–0.5 | | $\leq 10$ | <0.12 | SmartPlane Pro (https://www.drohnen.de/tag/tobyrich/) <br> DJI Mavic 3 (https://store.dji.com/de/product/dji-mavic-3) |
| Small | 0.51–2 | <9 | <185 | <0.4 | Wingcopter 198 (https://wingcopter.com/) <br> Astro (https://freeflysystems.com/alta-x) <br> Scorpion (https://www.quantum-systems.com/) |
| Medium | 5–10 | <200 | | <1.1 | Trinity F90+ (https://www.quantum-systems.com/) <br> Vector (https://www.quantum-systems.com/) <br> Alta X (https://freeflysystems.com/alta-x) <br> Yangda YD6-1600S (https://www.yangdaonline.com/) |
| Large | >10 | <600 | <463 | <5.5 | Primoco UAV One 150 (https://uav-stol.com/primoco-uav-one-150/) <br> Feng Ru 3-100 (https://ev.buaa.edu.cn/info/1133/3165.htm) |

## 4. Problem Statement

This section introduces several notations and formal definitions relevant to the problem statement. Following this, we will present a mathematical description of the problem of finding targets in a distributed environment. Overall, the use of a multi-UAV system to reach several targets in a large environment goes through two steps, as shown in Figure 5. First, the pre-processing step, which is performed on the GCS, consists in analyzing the properties of the environment in order to decide on the partitioning logic as well as the path planning algorithm. Then, in the second phase, each UAV assigned to its sub-environment is tasked with detecting, classifying, and finding the optimal path to reach a well-defined target in a distributed manner.

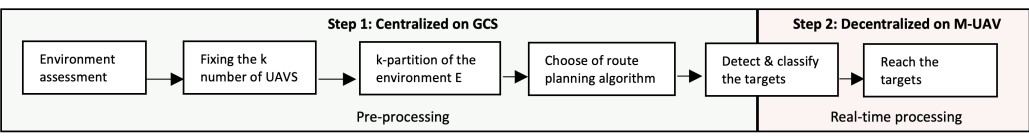

**Figure 5.** Workflow of target detection with multi-UAV system.

### 4.1. Notations and Formalism

#### 4.1.1. Environment E

An environment $E = (P, L, W)$ is a three-dimensional finite space composed of the following:

- A set of points $P = \{p_1, ..., p_n\} \rightarrow \mathbb{R}^3$ such as $\forall i \in [\![1..n]\!]$; the positions $p_i = (x_i, y_i.z_i) \in \mathbb{R}^3$ of these waypoints are defined by 3D Cartesian coordinates.

- A set of lines $L = \{(p_i, p_j) | p_i, p_j \in P \times P\}$ that connect the waypoints.
- A weighting function $W : E \rightarrow \mathbb{R}$ associated with the lines of $E$, such as $W = \{w(p_i, p_j) | p_i, p_j \in P^2 \text{ and } p_i \neq p_j\}$.

The number of finite elements of $E$ is the cardinal $|P|$ and the density (weight) of the environment is $|L|$. We will denote the shape of the environment by $s(E)$

### 4.1.2. Sub-Environment $E_i$

Let $E = (P, L, W)$ and $E_i = (P_i, L_i, W)$ be two environments. $E_i$ is a sub-environment of $E$ if $P_i$ is a subset ($P_i \subset P$) of $P$ and $L_i$ is a subset ($L_i \subset L$) of $L$.

Then, the shape and density of the sub-environment $E_i$ are, respectively, $s(E_i) \leq s(E)$ and $|L_i| \leq |L|$.

### 4.1.3. Partition $P_E^k$

Let $E$ be a non-empty environment and $k \geq 2$ is the number of sub-environments in the partition. We say that $P_E^k = \{E_1, ..., E_k\}$ is a k-partition of $E$ if

- $\forall i \in [\![1..k]\!], E_i \neq \emptyset$.
- $\forall i, j \in [\![1..k]\!]$ such as $i \neq j$; we obtain $E_i \cap E_j = \emptyset$.
- $\bigcup_{i=1}^{k} E_i = E$.

Thus, the elements of $P_E^k$ (sub-environments of $E$) are not empty and they are disjoint two by two. In other words, they have no element in common two by two.

The balance $B(P_E^k)$ indicates that the partitioning is spread equally. It is calculated as follows:

$$B(P_E^k) = \frac{\max\{|L_1|, \cdots, |L_k|\}}{|L_{avg}|} \tag{1}$$

where $|L_{avg}|$ is the average weight of the partition.

$$|L_{avg}| = \frac{\sum_{i=1}^{k} L_i}{k} \tag{2}$$

If the criterion $B(P_k) < 1 + \epsilon$ is met, then the partition $P_k$ is well balanced considering an acceptance error $\epsilon$.

### 4.1.4. Target $T$

A collection of finite elements $T = \{T_1, ..., T_k\}$ that lie in $E$ are defined by a class $C$ of discrete objects such that $E \times C \rightarrow T$. Here, $T_i$ is the set of targets associated with the sub-environment $E_i$, such as the following:

- $T = \emptyset$, if $E$ contains no target.
- $\bigcup_{i=1}^{k} T_i = T$.
- $\forall i, T_i \neq \emptyset$, if and only if $E_i$ contains at least one target.
- $\forall i, j \in [\![1..k]\!]$ such as $i \neq j$, we obtain $|T_i| > |T_j| \Rightarrow E_i$ and it contains more targets than $E_j$.
- $\forall i, j \in [\![1..k]\!]$ such as $i \neq j$, we obtain $T_i \cap T_j = \emptyset \Rightarrow$ and they share common targets.

### 4.1.5. M-UAV

M-UAV is a swarm of $k$ UAVs such as M-UAV = $\{\text{UAV}_1, ..., \text{UAV}_k\}$. We assume that we assign each drone or $\text{UAV}_i$ to its sub-environment $E_i$ in such a way that the environment $E$ is covered by M-UAV.

### 4.1.6. Perception $P$

Perception $P = (E, T, A)$ is the capability of the swarm M-UAV to take information related to $E$ as input and $T$ to perform a set of actions $A$ according to the function $f$ defined as $f : E \times T \mapsto A$.

$\forall i \in [\![1..k]\!]$, the perception $P_i$ of a given drone UAV$_i$ from its observation on $E_i$ and $T_i$ are defined as $P_i = (E_i, T_i, A)$.

### 4.2. Collaborative Target Detection Problem

The exploration of an environment $E$ by M-UAV consists of detecting, analyzing, and finding the path to reach these targets. For a large-scale environment, the complexity becomes exponential with respect to the number of these elements. This becomes even more problematic when using a UAV based on a centralized architecture (UAV-to-GCS) because of the latency. In addition, the UAVs' limited flying autonomy must be considered. Therefore, to cope with these problems, a decentralized M-UAV system based on the divide-and-conquer [56] technique should be used. It consists in dividing the exploration problem into sub-problems according to the number $k$ of drones used [12,40,49,57].

The division of the problem involves two challenges, which are (1) the partitioning of the environment $E$, commonly called the k-partition problem, and (2) the use of M-UAV such that each UAV$_i$ explores its sub-environment $E_i$ and computes the shortest path leading to a target of $T_i$; it is known as the Single-Source Shortest Path problem [58].

#### 4.2.1. k-Partition Problem

For a fixed $k$, the partitioning of a large environment $E$ consists in generating a k-partition $P_E^k$ such that $\forall i, j \in [\![1..k]\!]$, and we obtain $E_i \cap E_j = \varnothing$ while minimizing the number $|T_i \cap T_j|$ of shared targets between $E_i$ and $E_j$. A path is optimal if it minimizes the sum of the costs between two points. With the addition of obstacles and some constraints, this problem can change from a polynomial complexity to an NP-Complete problem [58]. This problem can be formulated as a linear optimization problem.

The first challenge is the minimization of shared targets between $E_i$ and $E_j$, as it reduces the communication costs between drones UAV$_i$ and UAV$_j$. The second challenge is balancing the weights of the $k$ sub-environments. For $k = 2$, it is a simple bi-partition of the environment, whereas for $k > 2$, the problem is NP-Complete [58] because there is no optimal solution algorithm in polynomial time.

Partitioning can be done in three ways, namely shape-based partition, density-based partition, and target-based partition, as illustrated in Figure 6. Shape-based partitioning consists of partitioning the environment $E$ into $k$ sub-environments with identical dimensions, such as $\forall i, j \in [\![1..k]\!]$, and we have $s(E_i) \approx s(E_j)$. On the other hand, the density-based partition consists in having a k-partition such as $\forall i, j \in [\![1..k]\!]$, and we have $|E_i| \approx |E_j|$. Finally, the one based on targets allows us to obtain a partition of the environment by considering the uniform distribution of targets on each sub-environment, such as $\forall i, j \in [\![1..k]\!]$, where we have $|T_i| \approx |E_j|$.

We conclude that the mathematical model associated with the partitioning of the environment $E$ within the constraints is described as follows:

$$\begin{cases} \underset{i \neq j \in [\![1..k]\!]}{\text{minimize}} & |T_i \cap T_j| \\ \text{subject to:} & B(P_E^k) \leq 1 + \epsilon \\ & E_i \cap E_j = \varnothing \\ & E_i \neq E_j \\ & \cup_{i=1}^{k} E_i = E \\ & \cup_{i=1}^{k} T_i = T \end{cases} \tag{3}$$

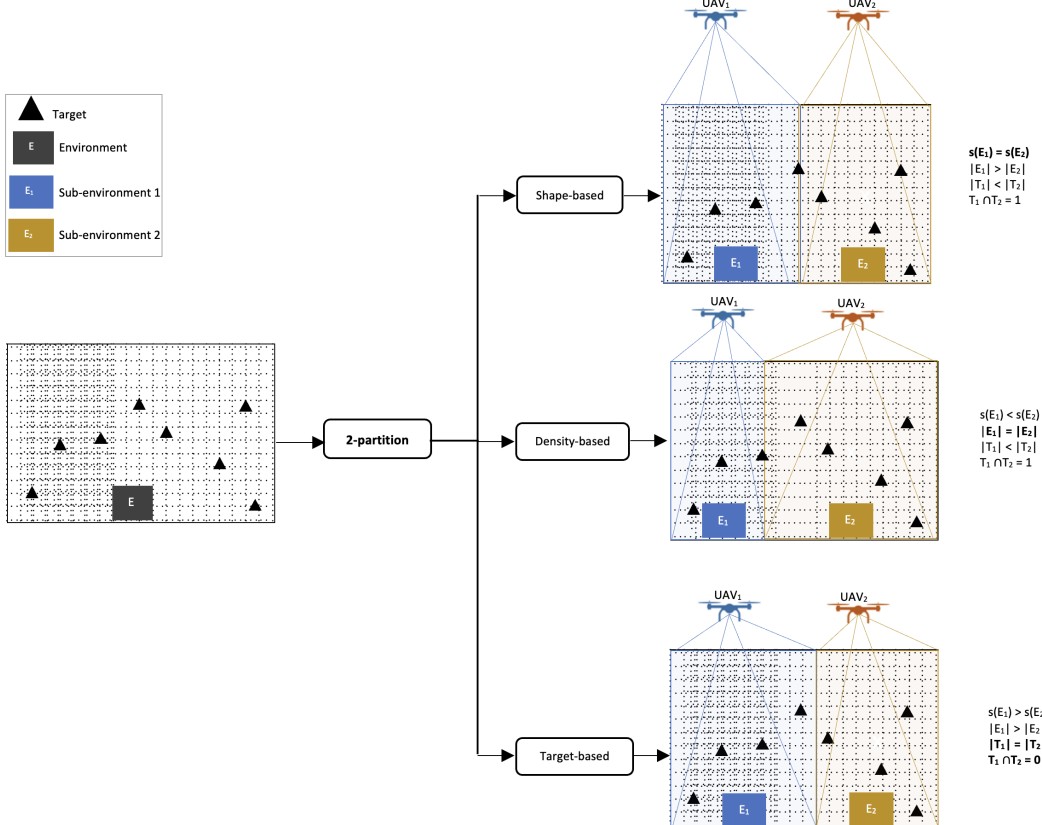

**Figure 6.** Illustration of a bi-partition ($k = 2$) of an environment for two UAVs with the three partitioning approaches.

#### 4.2.2. Single-Source Shortest Path Problem (SSSP)

After the partitioning of the environment $E$ into $P_E^k$, the next step consists in finding the paths that join a set of targets $T_i$ detected in $E_i$ in a distributed way. Each drone UAV$_i$ calculates the shortest path to reach a target at the position $p_{target}$ from its initial position $p_{init} \in V_i$, as shown in Figure 7. It is an algorithmic optimization problem referred to as the Single-Source Shortest Path problem and is well known in operations research and graph theory [59]. It consists in finding an optimal or near-optimal path between two nodes of a directed graph. It is a multi-criteria optimization problem aiming at satisfying the reachability of the final point $p_{target}$ while minimizing the costs (e.g., distance or time) of the path.

There are three variants of the SSSP problem: (1) the dual-path problem, which consists of finding a set of paths leading to a target from several sources; (2) the all-pair problem, which consists in finding the optimal path between all pairs of points; and, finally, (3) the single-source problem, which consists in finding an optimal path from a source that joins all the targets $T_i$ of a sub-environment $E_i$. In our study, we focus on the latter type of problem.

Suppose a drone UAV$_i$ assigned to the sub-environment $E_i = (P_i, L_i, W)$ for an exploration mission that consists in finding a set of targets $T_i$ such as the following:

- $p_{init} \in P_i$: the origin point or current position of the drone UAV$_i$.
- $p_{target} \in P_i \cap T_i$: the goal point or position of the target $t_i \in T_i$.
- $p_{init} \neq p_{target}$.
- path: the path to reach $p_{target}$ from $p_{init}$.

We consider the linear program by introducing a binary decision variable $x_{s,d} \in \{0, 1\}$. This variable indicates whether the weight $w(p_s, p_d)$ associated with the line between $p_s$ and $p_d$ minimizes the path cost. It is equal to 1 if the weight minimizes the cost $w(path)$ of the path and 0 in the opposite case:

$$\min w(path) = \sum_{(p_s, p_d) \in L_i} w(p_s, p_d) \times x_{s,d} \tag{4}$$

Thus, each $UAV_i$ will seek to determine the minimum weight lines that optimize and favor the progression towards the target $p_{target}$ position from its initial position $p_{init}$. This condition is defined as a flow conservation problem [60]; it stipulates that the sum of the flows into a given point must be equal to the sum of the flows out of that point $p_i$, apart from the start $p_{init}$ and end $p_{target}$ points.

The mathematical formulation of this condition is formulated as follows:

$$\forall p_i \in P_i \backslash \{p_{init}, p_{target}\}, \quad \sum_{(p_i, p_j) \in L_i} x_{i,j} = \sum_{(p_j, p_i) \in L_i} x_{j,i} \tag{5}$$

For each $UAV_i$, the mathematical model of the problem of reaching the position $p_{target}$ of a target $t \in T_i$ in its environment $E_i$ is described as follows:

$$\begin{cases} \underset{p_i \neq p_j \in P_i}{\text{minimize}} & \displaystyle\sum_{(p_i, p_j) \in L_i} w(p_i, p_j) \times x_{i,j} \\ \text{subject to:} & \displaystyle\sum_{(p_i, p_j) \in L_i} x_{i,j} = \sum_{(p_j, p_i) \in L_i} x_{j,i} \\ & x_{i,j} \in \{0, 1\} \\ & p_i, p_j \in P_i \backslash \{p_{init}, p_{target}\} \end{cases} \tag{6}$$

If we set $d(p_i)$ as the distance to reach the point $p_i$ from the initial point $p_{init}$ and if we set $d(p_{target}) = 0$, then, for each point $p_i$ to be explored by the $UAV_i$ in its environment $E_i$, the resulting mathematical model will be written as follows:

$$\begin{cases} \underset{p_i \neq p_j \in P_i}{\text{minimize}} & \displaystyle\sum_{p_i \in P_i} (d(p_{init}) - d(p_i)) = d(p_{target}) - d(p_{init}) \\ \text{subject to:} & d(p_j) - d(p_i) > w(p_i, p_j) \end{cases} \tag{7}$$

There are several shortest path search algorithms; the most popular are Bellman–Ford [61], Dijkstra [62], Floyd–Warshall [63], and A* [64,65], which uses a heuristic function to predict the path ahead and avoid obstacles.

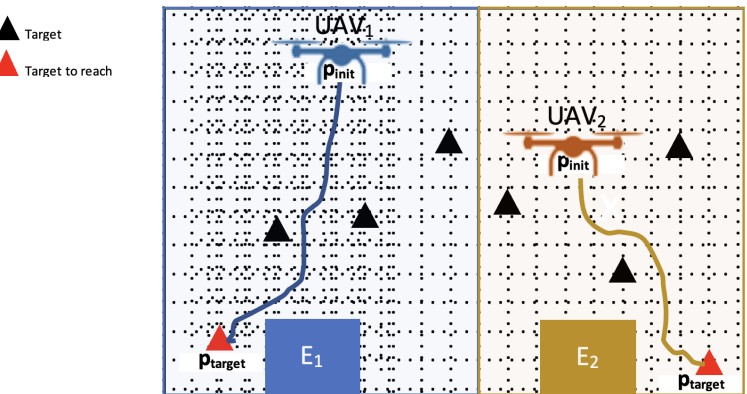

**Figure 7.** Illustration of a decentralized path on $UAV_1$ and $UAV_2$ to reach targets, respectively, in sub-environments $E_1$ and $E_2$.

## 5. State of the Art

The concept of multi-UAV was first used in the military domain for recognition, before being popularized in the industrial and civilian domains [66]. The fact of having several

UAVs has been shown to minimize the duration of missions and increase the satisfaction rate, especially for complex missions [27].

### 5.1. Military

The use of multi-UAV in the military field represents a real advantage, such as lowering the loss of both soldiers' lives and the military arsenal [2]. In October 2016, the US Department of Defense (https://www.defense.gov/News/Releases/Release/Article/1044 811/department-of-defense-announces-successful-micro-drone-demonstration/, accessed on 15 March 2023) successfully tested one of the greatest deployments of 103 small UAVs in California. The micro-UAVs flew together under an adaptive formation and each drone collaborated with the others by using one distributed brain to achieve their mission.

In the military area, swarm UAVs are generally used for attack, defense [3], and intelligence missions such as surveillance and reconnaissance [2]. The authors in [2,3] have shown that the swarm UAV attack is effective for offensive missions because it allows them to completely invade the enemy's defenses with a high probability of hitting several targets. This is the tactic of "quality by quantity". It consists of attacking simultaneously a target according to a defined formation. The most used formation is the simultaneous attack in a dispersed way. For example, if a swarm composed of 5 UAVs attacks a target simultaneously, the probability of shooting down one UAV is $\frac{1}{5}$, which is 20%. This probability evolves according to the scalability of the UAVs in the swarm.

Although UAVs show a good ability in attack missions, they can also be used for defensive or counter-offensive purposes against other enemy UAVs. Scharre [3] showed that the use of numerous miniature decoy UAVs could disrupt the mission of the attacking swarm by eliminating some enemy UAVs. This will have a considerable impact on the cooperation and coordination mechanism of the attacking swarm. It should be noted that the number of UAVs per swarm is very decisive regarding who would win.

### 5.2. Surveillance and Reconnaissance

Surveillance and reconnaissance are two of the most popular use cases in which both the military and civilian domains cooperate. The rapid emergence of connected objects and the standardization of those related to UAVs as well as artificial intelligence through computer vision has created an ecosystem favorable to the integration of several surveillance sensors, such as RGB and thermal cameras. Mohamed et al. [27] showed that UAV surveillance is more advantageous than the conventional surveillance technique based on fixed CCTV cameras during reconnaissance patrols. In addition to minimizing the risk of casualties, it allows for the rapid and wide exploration of surveillance areas.

Multi-UAV systems are more suitable for surveillance and reconnaissance over a large area because the scalability of the number of UAVs per swarm favors the coverage of large areas and intelligence in less time. Ali et al. [26] presented a path planning system for real-time surveillance employing cooperative multi-UAVs. Since the main challenge of this work is similar to the problem of cooperative motion planning, they used a swarm of UAVs that need to control, monitor, and periodically generate mosaic maps of their respective mission areas.

Similarly, Matej et al. [29] presented a surveillance mission planning based on a cooperative multi-UAV system for space coverage optimization to reach suitable sensing locations. The proposed solution works in two phases. Firstly, they used the Rapid Exploration Random Tree algorithm to quickly explore the space until the sensing locations were found. Then, they used Particle Swarm Optimization to maximize the gathered information about the reached sensing locations.

The fields of application of multi-UAV systems in surveillance are varied [30,67]. One of them [67] integrated the Oriented Line Segment Coverage Problem (OLSC) model to improve intra-swarm cooperation. Each drone is fitted with an onboard camera that provides large coverage of the space in order to detect and identify the size and direction of targets. The architecture of the proposed model is centered around a GCS that plans

the missions and a master UAV that controls and coordinates the movements of the slave UAVs to cover all the targets. Following the same work, Daniel Perez et al. [30] have developed a GCS for surveillance missions with multi-UAV. The proposed system is decentralized, allowing the dispatching of tasks to the UAVs. Each UAV displays in real time its visualization operation in the 3D environment. They demonstrated the reliability of the system by using two quadrotors equipped with cameras.

*5.3. Payload Carrying*

The carriage of payloads such as packages is another area of growing interest in the field of logistics and supply chain management. Several companies in the distribution and delivery sector are conducting research on the automation of their delivery services (healthcare delivery, food delivery, postal delivery) by multi-UAV systems [12,39]. Most of the proposed approaches are adapted mainly for collaborative payload carrying with a set of UAVs.

Daniel Mellinger et al. [13] presented cooperative grasping and transport using a swarm of quadrotors to move a heavy payload of known structure and weight in a three-dimensional space. Since each UAV knows its position, the task of transporting the payload is decentralized in such a way that each UAV controls its trajectory with respect to the payload stabilization laws. On the other hand, the velocity and the state estimation of the positions of the swarm towards the goal are centralized on the GCS.

Following the same approach, the authors in [15] addressed the problem of carrying heavy rigid payloads with a multi-UAV system composed of small quadrotors. Each UAV is equipped with a camera and Inertial Measurement Unit sensor. The proposed approach allows them to (1) independently control the motion of each UAV while maintaining the stability of the system regarding the inertial and (2) each UAV benefits from a cooperative scheme, allowing them to measure the positions of the other UAVs.

Robin and Raffaello proposed in [14] a method for carrying a flexible structure of payload in three-dimensional space by using a swarm of quadrotors rigidly attached to it. The proposed approach is comparable to previous work [13], except that the stabilization and estimation of the payload deformation are centralized on the GCS. They use a linear-quadratic regulator (LQR) function, which allows them to control and regulate the parameters governing the payload shape.

*5.4. Exploration*

The ability of UAVs to reach inaccessible areas is a great advantage for exploratory missions, especially in land and marine environments. Nowadays, UAVs are commonly used for geological exploration and archaeological excavation missions. In [68], the authors applied model-free deep reinforcement learning on a UAV swarm to accomplish exploration missions. Each UAV is defined as an autonomous agent that interacts with its environment through some actions. For each movement and action, the agent receives a reward (bonus or malus). The objective is to maximize the reward of the agents' actions with the aim of better understanding the exploration policy of the environment.

Singh et al. [69] addressed the challenge of multi-UAV exploration by optimizing the paths of UAVs investigating a large and dynamic environment. The optimal environment exploration challenge is described as a pre-learned Gaussian process and is solved centrally. Each UAV agent evaluates the quality of its exploration process by using mutual information between both the explored and the rest of the environment.

Furthermore, decentralized techniques provide well-known exploration benefits. In [70], the authors proposed a decentralized multi-agent version for environment exploration. They tackled the following three challenges: determining which information should be shared to enable unsupervised multi-agent coordination; (2) avoiding collisions; and (3) learning the environment's hyper-parameters while exploring.

Following the previous work, a decentralized control approach for exploring unknown environments with a multi-Turtlebot system composed of three robots was presented in [17].

For controlling the coordination between the uncrewed robots, they implemented a greedy decentralized method, allowing each robot to navigate and map an unknown environment without exceeding its boundaries.

There are also other research projects [8,9] that expand the exploration to archaeological excavation missions. For example, Efstathios and Fulvio [8] showed a state-of-the-art and deep analysis of archaeological remote sensing based on multi-UAV systems. Most existing methods are time-consuming, require a significant amount of hardware resources, and are computationally expensive. Using a multi-UAV system not only decreases the mapping time by distributing the coverage area, but it also enhances the map quality through information sharing [49]. It is possible to create a fast 3D map of archaeological sites using a collaborative mapping approach. This approach consists in partitioning the archaeological space and assigning portions to each UAV. Thus, each UAV performs the mapping task of its space and exchanges information on the boundaries of its space to the whole swarm for the achievement of the mapping mission in a distributed way [9].

*5.5. Other Applications*

The application domains of multi-UAV are varied and almost ubiquitous. Apart from the mentioned works, several use cases remain to be explored. Sharma et al. [48] present an application of multi-UAV systems for road traffic monitoring. There, they present a swarm coordinated with Vehicular Ad-Hoc Networks (VANETs) for vehicle tracking and driver behavior analysis. This approach reduces the number of accidents due to unsafe drivers and regulates traffic better. More applications and challenges of multi-UAV systems related to intelligent transportation systems for smart cities are highlighted in [16,71].

Another research work [72] in the same scope presents a cooperative traffic monitoring system based on a two-layer network (UAV aerial layer and vehicle ground layer). The UAVs in the FANET layer work together to control vehicles from the ground in the VANET layer. The link between UAVs is performed through air-to-air (A2A) communication, while the one inside the ground vehicular sub-network relies on air-to-ground (A2G) communication.

Another relevant use of multi-UAV is search and rescue [7,28]. The authors in [7] implemented a rescue mission using a set of stream videos from a swarm to a distributed ground control station. A similar work [28] introduces a machine learning approach to predict the best UAV swarm formation, allowing them to find the optimal path loss for the rescue.

## 6. Existing Multi-UAV Communication Architectures

The communication architecture of a multi-UAV system is a key factor in the structure of the network infrastructure and the interoperability between UAVs. It has undergone various improvements, moving from centralized to decentralized communications [1,19] (see Figure 8). Centralized architectures are essentially based on the multi-directional communication model between UAVs and GCS [6], while the decentralized architectures are based on various models, allowing us to guarantee the scalability of UAVs, the interoperability between UAVs, and the coordination of several tasks on the UAVs to achieve a common goal [45].

The multi-UAV communication architectures are classified into four types, centralized, single-group, multi-group, and multi-level architectures [1], as shown in Figure 8. In this section, we will discuss the different research works conducted toward the decentralization of multi-UAV communication architectures.

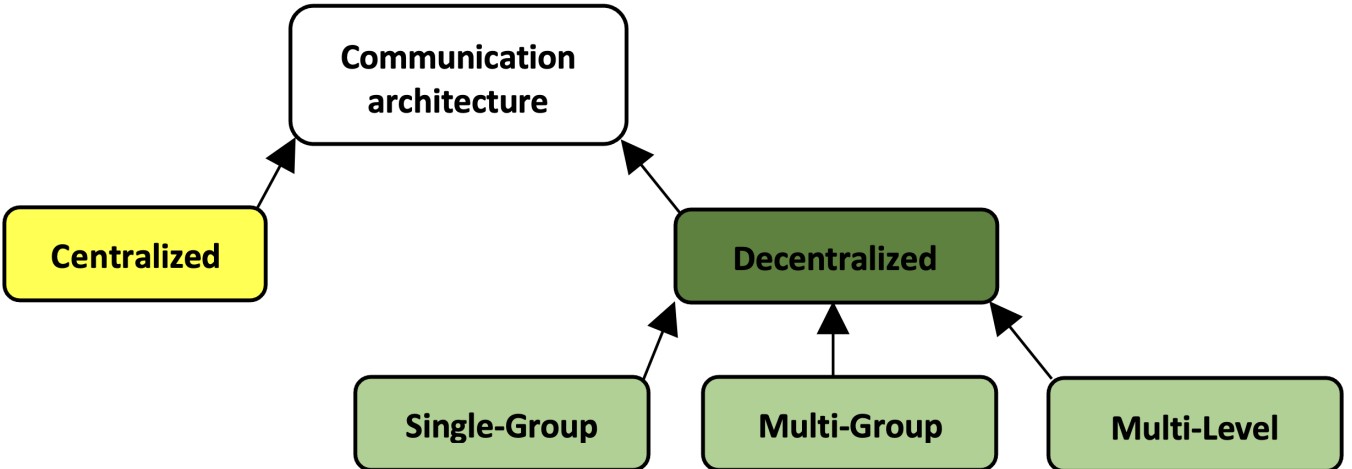

**Figure 8.** Types of multi-UAV communication architectures discussed in this paper.

*6.1. Centralized Communication Architecture*

A centralized multi-UAV architecture is a centralized data-oriented architecture that promotes only communication between UAVs and the GCS (air-to-ground) [6]. As shown in Figure 9, each UAV establishes directly two-way communication with the GCS. Moreover, the control and command operations are centralized in the GCS. Generally, this architecture supports mini-UAVs and implements simple routing protocols [73,74]. Its field of application is limited to straightforward missions with limited coverage.

However, this type of architecture does not support intra-UAV communications (air-to-air) [75]. Therefore, it is not possible to implement collaborative missions. Moreover, since there is a single point of failure in this architecture, it is vulnerable and can lead to system paralysis in case of failure [73]. In this context, the decentralization of the architecture will allow us to have intra-communication between the UAVs and to eliminate the dependencies on the GCS. The next section will cover the different types of decentralized architectures.

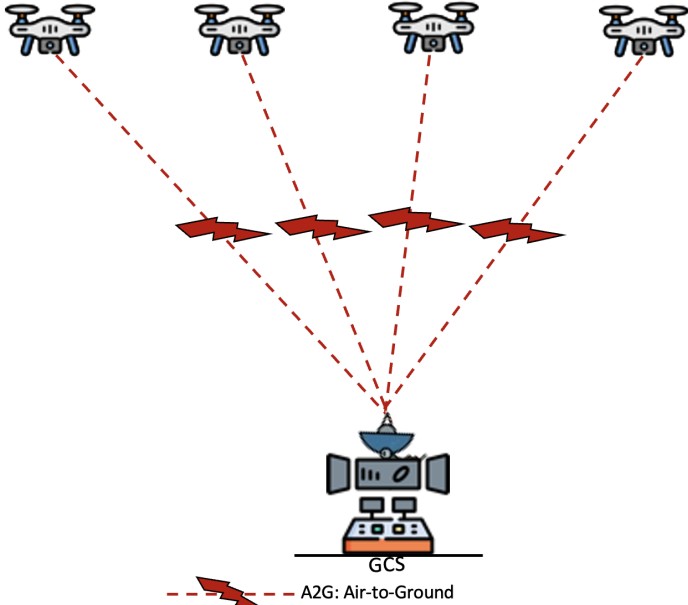

**Figure 9.** Multi-UAV communication architecture based on centralized framework.

*6.2. Single-Group Communication Architecture*

A single-group architecture is commonly used for collaborative missions in small coverage areas. It is a decentralized architecture that promotes intra-communication between UAVs (air-to-air), thus eliminating the dependency on the GCS, as shown in Figure 10. The communication between the GCS and the UAVs (air-to-ground) is achieved through a particular UAV called a "Gateway UAV" [1]. It oversees the analysis of the information from the GCS before sharing it with all the UAVs according to the ad hoc network topology. In [1], the authors present three types of intra-UAV communication topologies, ring, star, and meshed, as shown in Figure 11.

1.  **Ring**: in an intra-UAV communication-based ring topology, the UAVs communicate in a bidirectional loop in a closed network. To guarantee the high availability of the information, any UAV can be used as a gateway to relay the information between the GCS and the rest of the UAVs. On the other hand, it is difficult to maintain the scalability of the UAVs because of the network topology.

2.  **Star**: in a star topology, a single UAV relays information from the GCS infrastructure and shares it with the others. Any communication between two UAVs must pass through the gateway UAV. It is easy to maintain scalability. However, it does not guarantee fault tolerance because, if the gateway UAV fails, the whole system stops.

3.  **Meshed**: it takes advantage of the ring and star topologies. It allows us to guarantee both scalability and fault tolerance.

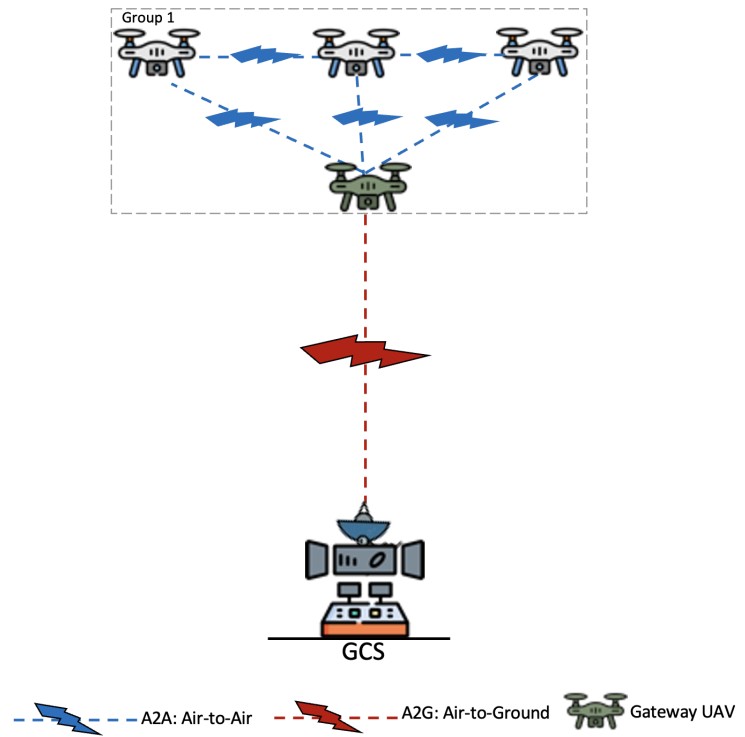

**Figure 10.** Multi-UAV communication architecture based on single group.

The diversified applications of UAVs have led to the increasing use of a variety of UAVs of different sizes and functions. Unfortunately, this architecture does not meet this type of demand because it does not allow the incorporation of different types of UAVs [1]. Moreover, it is adapted for short-range intra-communication, but it may be the case that some UAVs move away from the others during certain missions in large environments. To overcome this problem, an alternative is to create several groups of UAVs. In the following, we will discuss the two types of multi-group architectures.

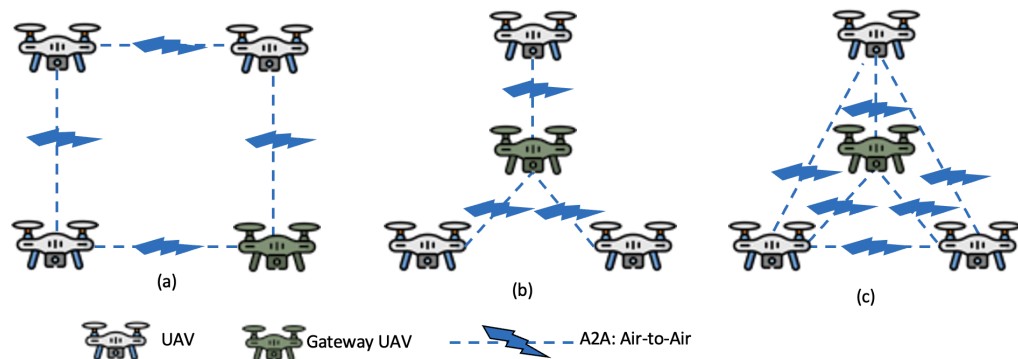

**Figure 11.** Types of intra-UAV communication: (**a**) ring topology, (**b**) star topology, (**c**) meshed topology.

### 6.3. Multi-Group Communication Architecture

A multi-group architecture takes advantage of both centralized and single-group architectures. As shown in Figure 12, it extends the single-group architecture into several groups of homogeneous UAVs that communicate with each other according to the intra-UAV topology [19] seen in Section 6.2. Each group has a gateway UAV. In addition, the communication between groups of UAVs (group-to-group) is done through gateway UAVs that communicate bidirectionally with the GCS. The robustness of this architecture favors collaborative missions over large coverage areas while maintaining control from the GCS [28]. However, there are some limitations to be mentioned:

- **Single point of failure**: if the gateway UAV fails, then all other UAVs in its group remain inactive. If the GCS fails, then the system will become totally paralyzed.
- **Homogeneity of UAVs**: in a group, the UAVs must be of the same type, and this is a hindrance for some missions. Moreover, this implies serious interoperability problems between groups of UAVs.

In the next section, we will discuss a type of architecture that overcomes the limitations of the multi-group architecture.

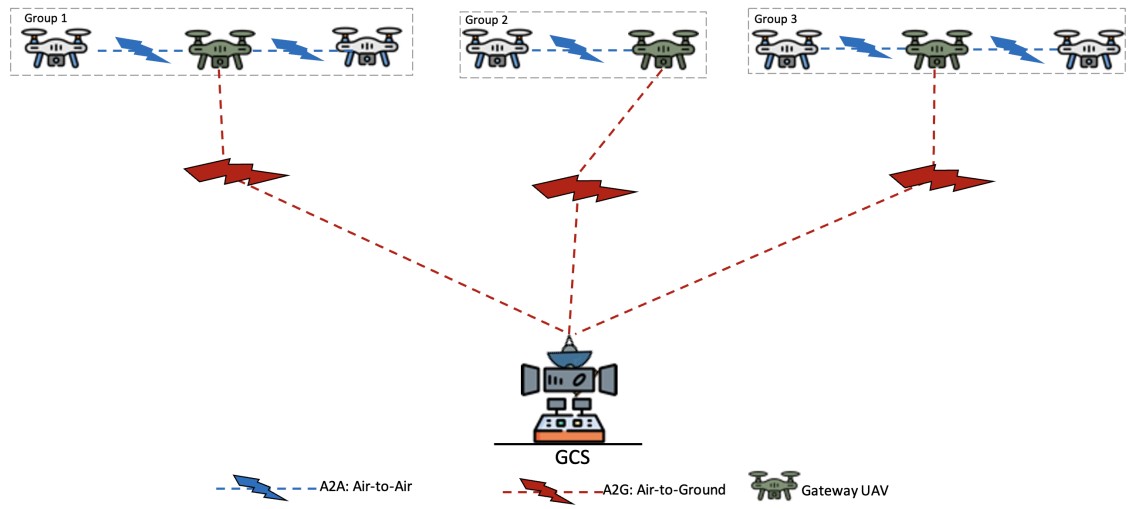

**Figure 12.** Multi-UAV communication architecture based on multi-group.

### 6.4. Multi-Level Communication Architecture

A multi-layer communication architecture is an extension of the multi-group architecture to a three-tier architecture. It is adapted to support the scalability of UAVs [19]. Unlike the multi-group architecture, the communication between the groups of UAVs is

achieved without using a GCS infrastructure [28]. It is composed of three layers, as shown in Figure 13:

1. **Layer 1 (A2A)** is adapted for UAV-to-UAV communication. It allows the intra-communication of a group of adjacent UAVs according to the three network topologies seen in Section 6.2.
2. **Layer 2 (G2G)** is suitable for group-to-group communication. It is responsible for the inter-communication of groups of UAVs. Each group of UAVs uses its gateway UAV to communicate with the adjacent groups. Moreover, this communication is done directly, without using the GCS network infrastructure.
3. **Layer 3 (G2A)** is suited to ground-to-air communication. It allows us to relay bidirectional information between the GCS and the UAV groups. Generally, this is done through the UAV gateway closest to the GCS infrastructure.

Compared to the three other architectures, the multi-level architecture is robust and supports fault tolerance. It is used for complex missions that involve different types of UAVs. In fact, the multi-layer structure of this architecture allows for the massive deployment of UAVs to cover a large airspace. Unlike the multi-group architecture, UAVs with different types of protocols can be deployed in the same group [19,28].

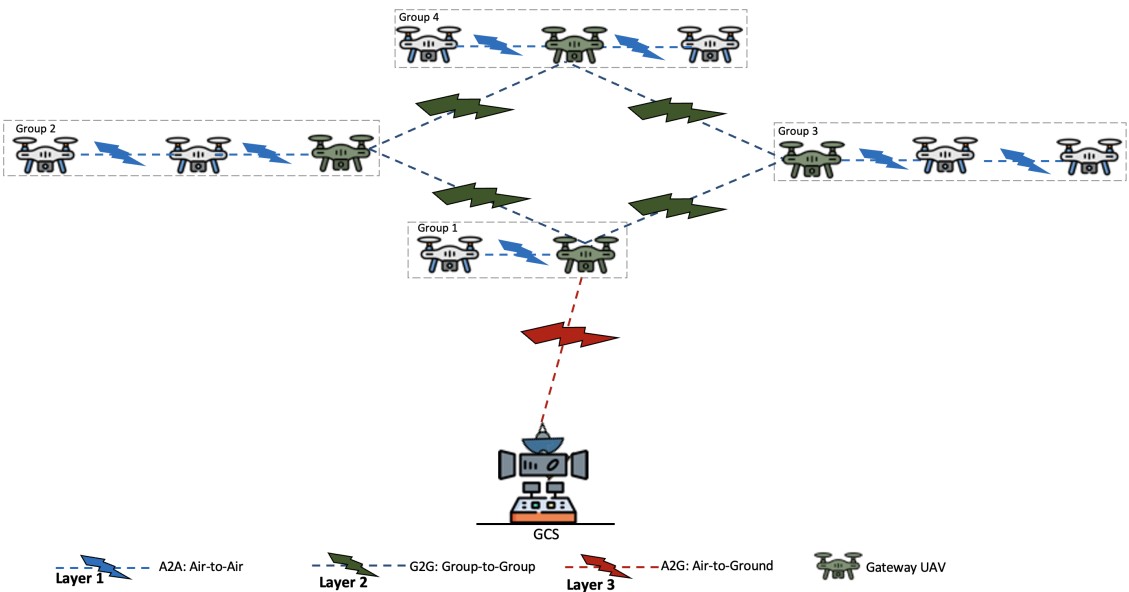

**Figure 13.** Multi-UAV communication architecture based on multi-level.

## 7. Comparison and Roadmap of Multi-UAV Architectures

The rapid emergence of UAVs and the diversity of their applications has led to a variety of communication architectures. Each one has its advantages and disadvantages. Table 3 summarizes the key performance of the four communication architectures seen in this paper. Multi-level and multi-group architectures are the major players. These are interoperable systems that support various types of UAVs. Their communication range extends over long distances, which allows them to ensure permanent connectivity in the network. They are also suitable for complex missions involving several types of UAVs that collaborate in real time. The robustness of these two architectures guarantees the scalability, fault tolerance, and stability of the system. They are currently implemented in the military domain for coordinated attack and defense missions [2,3,66].

The single-group architecture is suitable for small and medium UAVs. It is not interoperable and therefore does not allow the deployment of various types of UAVs. Its communication range is relatively dependent on the distance between UAVs. The connectivity becomes unstable when the UAVs move away from each other. It is appropriate for simple collaborative missions in limited environments. Bidirectional communication

(air-to-ground) between the GCS and the group of UAVs is achieved through the gateway UAV. When the gateway fails, the system shuts down. Most of the projects [13–15,57] related to the carrying of payloads implement this architecture since it offers peer-to-peer communication.

Finally, the centralized architecture is less efficient than the other three. It also does not support interoperability and its communication scheme is centralized on the GCS. This leads to latency in air-to-ground communications. Intra-UAV communication is also not supported in this type of architecture. Therefore, it is not suitable for collaborative mission planning. The only point of failure is the GCS: if it fails, then the whole system is paralyzed. Generally speaking, this architecture is used for very basic missions with relatively small UAVs [26,27,48,57].

**Table 3.** Key features of multi-UAV communication architectures discussed in this paper.

| Features | Centralized | Single-Group | Multi-Group | Multi-Level |
|---|:---:|:---:|:---:|:---:|
| Very small UAV | ☑ | ☑ | ☑ | ☑ |
| Small UAV | ☑ | ☑ | ☑ | ☑ |
| Medium UAV | ✖ | ☑ | ☑ | ☑ |
| Large UAV | ✖ | ✖ | ☑ | ☑ |
| Gateway UAV | ✖ | ✖ | ☑ | ☑ |
| Single-hop wireless | ☑ | ☑ | ☑ | ☑ |
| Multi-hop wireless | ✖ | ☑ | ☑ | ☑ |
| Interoperability | ✖ | ✖ | ☑ | ☑ |
| Stability | ✖ | ✖ | ☑ | ☑ |
| Fault tolerance | ✖ | ✖ | ☑ | ☑ |
| Cooperative mission | ✖ | ☑ | ☑ | ☑ |
| Streaming | ✖ | ☑ | ☑ | ☑ |
| Large aerial coverage | ✖ | ✖ | ☑ | ☑ |
| Air-to-air | ✖ | ☑ | ☑ | ☑ |
| Air-to-ground | ☑ | ☑ | ☑ | ☑ |
| Group-to-group | ✖ | ✖ | ☑ | ☑ |

The choice of an architecture is a delicate task during the conceptual phase of a multi-UAV system because it depends on the type and size of the UAVs, the area of the environment to explore, and the complexity of the mission to plan. If the choice is made out of context, a robust architecture could be inefficient on some types of UAVs and vice versa. Therefore, it is very important to define the type of mission and then fix the evaluation parameters seen in Table 3 before selecting an appropriate architecture.

Figure 14 shows a roadmap for multi-criteria decision support. It provides information on how to choose an architecture that is well adapted to the context. For simple missions including very small and small UAVs, it is preferable to opt for classical architectures. Ideally, if all operations are centralized on the GCS, then it is advisable to deploy a centralized architecture. If the mission involves sharing resources or exchanging information between UAVs, it is better to opt for a single-group architecture. It is important to ensure that each UAV remains within range of adjacent UAVs to maintain connectivity in the network. For complex missions, the ideal would be to use decentralized architectures. In this sense, multi-group and multi-layer architectures remain the best choices. The robustness of the multi-level architecture makes it an appropriate choice for mission planning involving several different tasks [57].

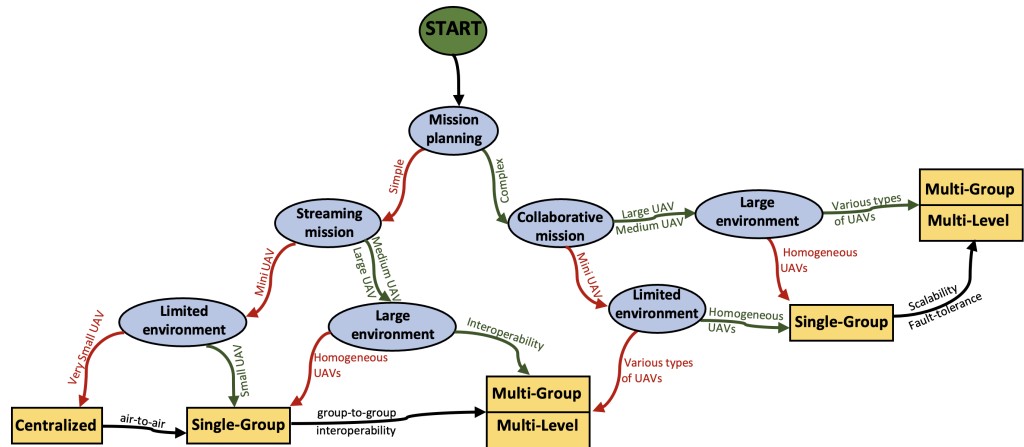

**Figure 14.** Roadmap of multi-UAV communication architectures.

## 8. Routing Protocols of Multi-UAV Systems

The increasing number and variety of mobile connected objects has led to numerous studies [22,76] to develop routing protocols adapted to highly mobile environments. Each protocol is adapted according to a particular routing standard, which is largely influenced by the communication infrastructures. A protocol fits into the acceptance standards if it allows us to convey, ensure the integrity of, and secure packets (messages) sent between the nodes of its network. In the case of UAV networks, the communication protocols used are adapted for Flying Ad-Hoc Network (FANET) routing protocols [19]. In this paper, we investigate 17 FANET routing protocols. We distinguish three categories of routing protocols: topology [23,75], geographic position [20,77,78], and hierarchical-based [79,80] routing protocols; refer to Figure 15.

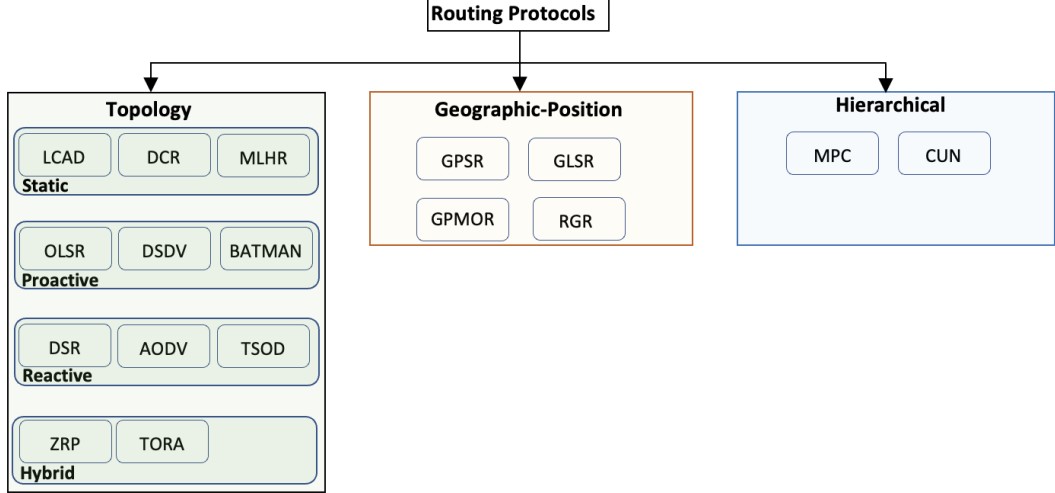

**Figure 15.** Classification of FANET routing protocols discussed in this paper.

### 8.1. Topology-Based Routing Protocols

Topology-based protocols [23,75] are classic routing protocols that employ a routing table that contains all the data transfer rules on a network; each of them is identified by its IP address (IPv4 or IPv6). For two UAVs in the network, the routing algorithm computes the appropriate route to send and receive packets between them based on the rules defined in the routing table. As shown in Figures 15 and 16, topology-based protocols are divided into four sub-categories: static [73,81,82], proactive [83,84], reactive [23,85–87], and hybrid [88] routing protocols.

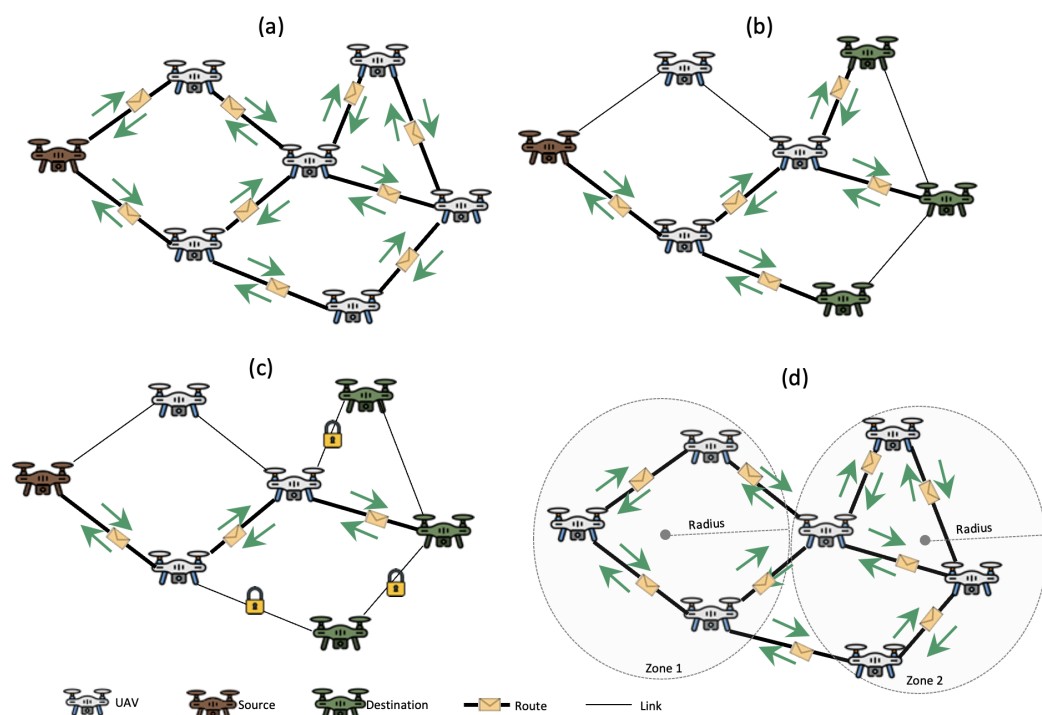

**Figure 16.** Paradigm of topology-based protocols: (**a**) static; (**b**) proactive; (**c**) reactive; (**d**) hybrid.

### 8.1.1. Static

Static-based routing protocols [73,81,82] are a subcategory of topology-based protocols that use a preconfigured static routing table for data transfer between UAVs referenced by their static IP addresses. To add a new UAV, the administrator must manually reconfigure the rules in the routing table to fit the scalability of the multi-UAV system. Therefore, this type of protocol is not suitable for a high-mobility network because of the lack of scalability and fault tolerance. The three variants of this sub-category of protocols are Load Carry And Deliver (LCAD) [73], Data-Centric Routing (DCR) [81], and Multi-Level Hierarchical Routing (MLHR) [82].

The LCAD protocol [73] is based on the Store-Carry-Forward mechanism [74]. It is adapted for a centralized architecture because it allows the transport of messages between an infrastructure (e.g., ground control station) and all UAVs. The communication mechanism of this protocol consists of three stages: (1) the load stage, in which the packets are stored in the UAVs; (2) the carry stage, in which the UAVs transmit these packets via the gateway UAVs; (3) finally, in the Deliver stage, the gateways deliver the packets to the final UAVs.

The DCR protocol [81] is based on a broadcasting mechanism. It is suitable for a single-group architecture. The transmission mechanism is based on the header attributes contained in the data. The data are sent only to the UAVs that want them, thus optimizing the network throughput. Then, the MLHR protocol [82], which is based on a node clustering paradigm, allows us to address the scalability challenge. The UAVs are hierarchically divided into clusters according to their types and roles. Each cluster contains a gateway UAV, which is responsible for sharing data with other clusters. Therefore, this protocol is well suited for a multi-group architecture.

### 8.1.2. Proactive

Proactive protocols [83,84] use dynamic routing tables. Each UAV has a routing table that stores, periodically for each UAV, the routes to broadcast messages to the destination UAVs. The delay between sending and receiving the message is very short because the paths are already known beforehand. However, when the swarm reaches a certain level of scalability, a network bottleneck ensues, as changes in the routing table are bandwidth- and

energy-intensive. For this reason, this protocol is not used for applications requiring a large number of highly mobile UAVs. The most well-known proactive protocols are Optimized Link State Routing (OLSR) [83], Destination Sequence Distance Vector (DSDV) [84], and Better Approach to Mobile Ad-Hoc Network (BATMAN) [89].

The OLSR protocol uses Dijkstra's algorithm [62] to calculate the routes for sending messages between the different UAVs, while the DSDV protocol uses the Bellman–Ford algorithm [61]. On the other hand, the BATMAN protocol optimizes the calculation of routes between UAVs. Unlike the OLSR and DSDV protocols, which calculate routes for all possible sending combinations, the BATMAN protocol only calculates routes for UAVs that are directly or indirectly (via the gateway UAVs) connected to each other.

### 8.1.3. Reactive

Reactive protocols [23,85–87] are optimized versions of proactive protocols that aim to minimize the bandwidth consumption. This sub-category of protocols is based on on-demand routing protocols, because routes are created only when necessary. If there are no data to share between two UAVs, then there is no need to calculate the route. This approach results in the efficient use of the bandwidth and reduced overhead. The most commonly used reactive protocols are Dynamic Source Routing (DSR) [85], Ad Hod On-demand Vector (AODV) [86], and Time-Slotted On-Demand (TSOD) [87].

DSR [85] is a dynamic source-routing protocol whose routing is initiated by the source UAV that wants to send a message. The complete route to transmit a message is indicated in the routing table of the source UAV. However, it is not possible to send messages if the route to the destination UAV is not registered in the header of the source UAV. The AODV protocol [86] leverages this limitation; it is the most used routing protocol in both VANET and FANET. Before sending a message, the source UAV checks if the routing path already exists. If so, it uses the existing route. Otherwise, it triggers a process to compute the route to the destination UAV.

On the other hand, a risk of collision between packets can occur when there are enough exchanges in the network. TSOD [86] allows us to avoid collisions; it is an extended version of AODV that introduces a time slot routing protocol. Each source UAV has a time slot allocated to it for sending messages, and they can only send messages during this slot. This protocol reduces significantly the lost packets and minimizes network bottlenecks.

### 8.1.4. Hybrid

This sub-category of protocols [88,90] is both proactive and reactive and is suitable for large mobile networks. It takes advantage of both to overcome the problems of bandwidth saturation, long delays, and collisions. Its operating principle consists in dividing the network of UAVs into sub-networks, commonly called zones. Intra-zone communications are based on the proactive protocols, while inter-zone communications are handled by the reactive protocols. The most popular hybrid protocols are the Zone Routing Protocol (ZRP) [90] and Temporarily Ordered Routing Algorithm (TORA) [88].

ZRP [90] divides the network into several zones of relatively equal radius and uses a dynamic routing table for the exchanges within the zones. Messages exchanged between UAVs from different zones are based on on-demand routing protocols. UAVs should always stay within the radius of their zones in order to maintain network stability. To address this limitation, researchers [88] have proposed TORA. TORA is a distributed version of a hybrid protocol that, instead of dividing the network, distributes the routing table in such a way that each UAV contains only information about its neighbors, rather than the entire network.

### 8.2. Geographic Position-Based Routing Protocols

Sometimes called georouting, these types of protocols rely on the geographical location services of the UAVs [20]. They are more adapted for highly mobile UAVs in large FANETs [19]. In this type of protocol, each UAV determines its GPS position and shares it

with its neighbors. A sender UAV must know the position of the receiver UAV, as shown in Figure 17. Based on this knowledge, messages can be routed without knowing the network topology beforehand. The paradigm of these protocols is similar to reactive protocols since the routes are used only when necessary. These protocols combine geoservice techniques [20,77] with the Greedy Forwarding approach [78] and they consist of Greedy Perimeter Stateless Routing (GPSR) [91], Geographic Load Share Routing (GLSR) [92], Geographic Position Mobility-Oriented Routing (GPMOR) [93], and Reactive Greedy Reactive (RGR) [78].

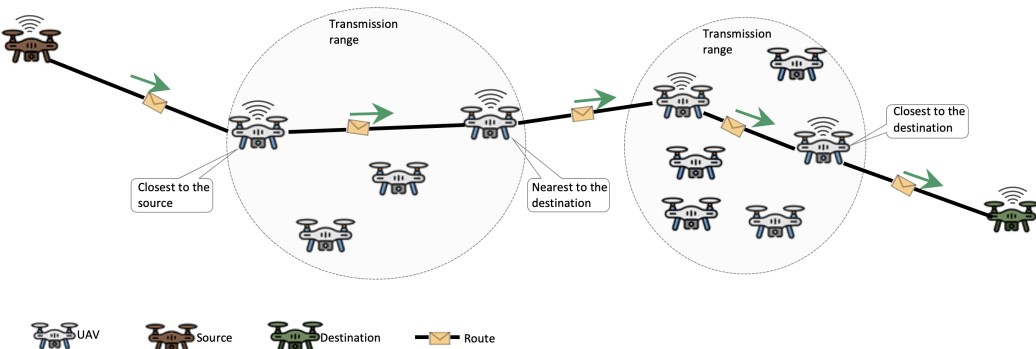

**Figure 17.** Geographic position routing paradigm.

GPSR was originally proposed for MANET and allows us to calculate the shortest route for sending messages. From the location information of each UAV, it uses the greedy method [94] to find the nearest neighbors of the UAV that receives the message. Then, it calculates the shortest route to send the message through one of the closest neighbors of the receiver UAV. If no route is found, then the sending fails, without any possibility to recover the message.

Cardellini et al. [92] proposed GLSR, an extended version of GPSR that uses a prediction function to select the next promising gateway UAV to compute multiple routes from the sender to the receiver UAV. In another work [93], the authors proposed GPMOR, an adapted version of GLSR for a highly mobile flying ad-hoc network. The proposed protocol uses Gauss–Markov stochastic processes to model and predict the movement of the UAVs. This approach makes it possible to anticipate the most optimal routes for sending messages.

To increase the deliverability of the messages and minimize the bandwidth consumption, Shirani et al. [78] introduced the RGR protocol. This protocol is based on both the AODV routing protocol and greedy forwarding method. Since it is a reactive protocol, it calculates "on-demand" only the routes for the UAVs waiting for the messages.

### 8.3. Hierarchical-Based Routing Protocols

In these types of protocols [79,80], the network is divided hierarchically into clusters. Each cluster is used for a specific function and has a head that facilitates inter-cluster communication through the other heads, see Figure 18. The communication process is supported by proactive routing protocols, while changes in the network or failures are managed by reactive protocol processes. The most popular of these routing protocols are Mobility Prediction Clustering (MPC) [79] and the Clustering Algorithm for Networking (CAN) [80].

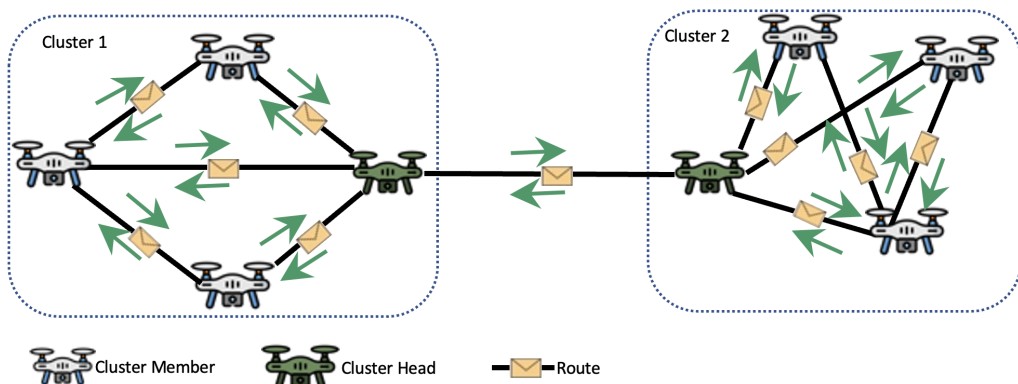

**Figure 18.** Hierarchical routing paradigm.

CNU is a clustering-based protocol that divides the network into clusters starting with the hierarchical level closest to the GCS. Thus, in an iterative way, new clusters are formed from the previous level's clusters until they cover the entire network. For each cluster, UAVs elect a cluster head based on their power law distributions in the graph network. The cluster head will be responsible for calculating the route to effectively broadcast messages inside the cluster. Thereafter, it shares them with other clusters through the cluster heads.

MPC is an adapted version of CAN for highly mobile FANETs. The MPC protocol uses mainly the Doppler shift method to compute the velocity and movements of a couple of UAVs. The clustering is done according to the state of each UAV: orphan (stateless), member (cluster member), or head (cluster head). Initially, all UAVs are in the orphan state. Afterwards, each UAV sends a "HELLO" message containing, in its header, the information relative to its velocity, direction, and identifiers. Depending on the change in wave frequency (obtained from Doppler shift) between the transmitter UAV that moves compared to the receiver UAV, a state (member or head) is assigned to each UAV until the clusters are completely formed.

## 9. Comparison and Roadmap of Routing Protocols

As we have seen in the previous sections, UAVs are varied from very small to large. They are either adapted to be deployed with a traditional or a decentralized communication architecture. To this, we must add the interoperability between UAVs, which is mainly based on routing protocols. Thus, each protocol has its benefits and drawbacks. Table 4 summarizes the key performance of the FANET routing protocols reviewed in this paper. We note that topology-based routing protocols are only suitable for centralized and single-group architectures. They do not provide routing for highly mobile networks. It should also be noted that only hybrid and BATMAN protocols are suitable for sending messages without a risk of collision or packet loss. OLSR and DSDV protocols are much more expensive in terms of bandwidth, even if they are fast and avoid redundancy during message transmission. As for geographic position-based protocols, they operate on centralized, single-group, and multi-group architectures. These types of protocols promote interoperability and ensure good message deliverability while reducing bandwidth throughput. On the other hand, they do not guarantee fault tolerance in case of the failure or inaccessibility (wrong GPS coordinates) of a UAV. Only the GPSR and GLSR protocols are suitable for communication in a large-scale network. Finally, only the hierarchical-based routing protocols are operational on the four types of architectures. They are suitable for collaborative missions that involve frequent data exchange over several hierarchical levels of the network. Although they are efficient, some limitations should be noted. (1) The MPC protocol requires more time to send a message, which largely impacts the actions of other UAVs if they depend on it. (2) In case of message loss, the CAN protocol does not ensure its recovery, which can lead to the paralysis of the mission and cause a network bottleneck.

**Table 4.** Key features of FANET routing protocols discussed in this paper.

| Protocols | | Architectures | | | | Performance Metrics | | | | | | | | |
|---|---|---|---|---|---|---|---|---|---|---|---|---|---|---|
| | | Centralized | Single-Group | Multi-Group | Multi-Level | Deliverability | Collision | Scalability | Delay | Throughput | Redundancy | High-Mobility | Link Failure | Interoperability |
| Topology | LCAD | ✓ | ✗ | ✗ | ✗ | ✓ | ✗ | ✗ | ✗ | ✓ | ✓ | ✗ | ✗ | ✗ |
| | DCR | ✓ | ✓ | ✗ | ✗ | ✓ | ✗ | ✗ | ✓ | ✓ | ✗ | ✗ | ✗ | ✗ |
| | MLHR | ✓ | ✓ | ✓ | ✗ | ✓ | ✗ | ✓ | ✓ | ✓ | ✗ | ✗ | ✗ | ✓ |
| | OLSR | ✓ | ✓ | ✗ | ✗ | ✗ | ✗ | ✗ | ✓ | ✗ | ✓ | ✗ | ✓ | ✗ |
| | DSDV | ✓ | ✓ | ✗ | ✗ | ✗ | ✗ | ✗ | ✓ | ✗ | ✓ | ✗ | ✗ | ✗ |
| | BATMAN | ✓ | ✓ | ✗ | ✗ | ✓ | ✓ | ✓ | ✓ | ✓ | ✓ | ✗ | ✓ | ✗ |
| | DSR | ✓ | ✓ | ✓ | ✓ | ✗ | ✗ | ✓ | ✗ | ✓ | ✓ | ✗ | ✗ | ✓ |
| | AODV | ✓ | ✓ | ✗ | ✗ | ✓ | ✗ | ✗ | ✗ | ✓ | ✓ | ✗ | ✓ | ✗ |
| | TSOD | ✓ | ✓ | ✗ | ✗ | ✓ | ✓ | ✗ | ✗ | ✓ | ✓ | ✗ | ✓ | ✗ |
| | ZRP | ✓ | ✓ | ✓ | ✗ | ✓ | ✓ | ✓ | ✓ | ✓ | ✓ | ✗ | ✓ | ✓ |
| | TORA | ✓ | ✓ | ✓ | ✗ | ✓ | ✓ | ✓ | ✓ | ✓ | ✓ | ✓ | ✓ | ✓ |
| Geographic position | GPSR | ✓ | ✓ | ✓ | ✗ | ✓ | ✓ | ✓ | ✗ | ✓ | ✓ | ✓ | ✗ | ✓ |
| | GLSR | ✓ | ✓ | ✓ | ✗ | ✓ | ✗ | ✓ | ✓ | ✓ | ✗ | ✓ | ✗ | ✓ |
| | GPMOR | ✓ | ✓ | ✓ | ✗ | ✓ | ✗ | ✗ | ✓ | ✓ | ✗ | ✓ | ✗ | ✓ |
| | RGR | ✓ | ✓ | ✓ | ✗ | ✓ | ✗ | ✗ | ✓ | ✓ | ✗ | ✗ | ✗ | ✓ |
| Hierarchical | CAN | ✓ | ✓ | ✓ | ✓ | ✓ | ✓ | ✓ | ✓ | ✓ | ✓ | ✓ | ✗ | ✓ |
| | MPC | ✓ | ✓ | ✓ | ✓ | ✓ | ✓ | ✓ | ✗ | ✓ | ✓ | ✓ | ✓ | ✓ |

The choice of routing protocol depends on the type of communication architecture, the mission requirements, and the relevant features seen in Table 3. It is very important to analyze this step thoroughly in order to avoid problems of incompatibility between the protocols and the communication architecture. Figure 19 shows how to choose the right protocol depending on the infrastructure and the communication architecture. For a centralized architecture, it is better to opt for topology-based protocols. If the deliverability of messages is an important factor, then it is better to use static protocols. In case interactions with as few collisions as possible are required, it is more advantageous to use hybrid protocols. For decentralized architectures involving highly mobile UAVs in a large-scale environment, it is better to use geo-based protocols. In contrast, if it is a collaborative mission, hierarchical-based routing protocols are ideal.

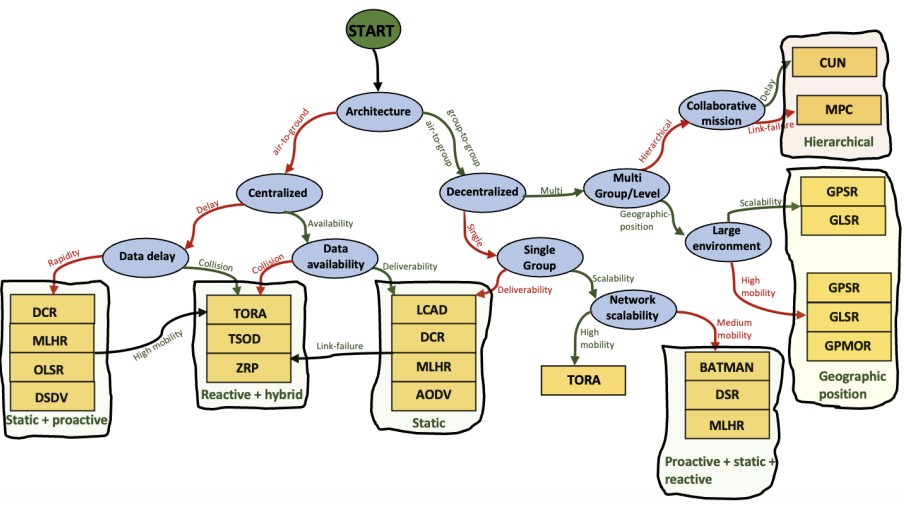

**Figure 19.** Roadmap of FANET routing protocols.

## 10. Open Challenges and Future Directions

The deployment of a multi-UAV system involves several challenges to overcome, ranging from the choice of infrastructure to the planning of the mission. Despite efforts in the field, most research work is still in the prototyping or experimental phase for some. Several challenges remain to be met to effectively exploit swarms in upcoming trends. These topics include the collaborative decision making, the communication infrastructure, the data streaming, and the regulation of the use of combined UAVs.

### 10.1. Collaborative Decision Making

The perception of the environment, the states, the allocation and coordination of tasks, as well as the consensus are important factors that influence the behavior of UAVs in a swarm. Each autonomous UAV contributes to the mission progress by performing actions in a collaborative manner. In this context, the level of human control is reduced to a few start-up operations or maintenance operations.

Several challenges remain to be addressed, including consensus building, task allocation, collective failure management, and perception.

- **Definition of a consensus** on which the UAVs base themselves to carry out individual actions that contribute to a convergence towards a common goal.
- **Task allocation**, which consists in dynamically assigning tasks according to the state and capacity of the UAVs. To allow a high level of parallelism, tasks can be distributed fairly if the UAVs are homogeneous. Otherwise, it will be done according to the workload and computing resources of each UAV. It should also be noted that the assignment of tasks is based on a parallelism paradigm (synchronous parallelism, asynchronous parallelism, or bulk synchronous parallelism) that ensures that the tasks are executed concurrently.
- **Collective failure detection** in the swarm makes it possible to detect the UAVs that have abnormal behavior, or that carry out actions that do not contribute to the realization of the mission. For example, due to a hardware malfunction (e.g., low battery or defective rotor), a UAV may not respond appropriately to some tasks assigned to it. The collective management of the failures in the swarm must make it possible to detect the UAVs whose behavior does not contribute to the progress of the mission. It is also necessary to consider an activity continuity plan that allows the other UAVs to continue the mission and to adapt to any type of change.
- **Perception of the environment** represents the information collected by the UAV. Each UAV uses onboard sensors to build a local representation of the environment. The combined set of perceptions allows the swarm to build a global representation and make more predictive decisions. For example, to calculate the most optimal path that reaches a target, we can choose the closest drone. See Section 4.2.2 for more details.

These challenges open future directions as follows:

- **The deployment of a swarm using a multi-agent-based model** for better coordinating the actions and interactions of the drones to enhance the individual and/or collective decision making.
- **The implementation of a reinforcement learning algorithm** on each drone to control its behavior based on the reward of good actions.

### 10.2. Communication Infrastructure

The existence of several communication architectures, the variety of routing protocols, and the variety of UAV types raise the question of the choice of an adapted communication infrastructure. This issue higlights the challenges of interoperability in a distributed manner [95]. Generally, in a decentralized architecture, UAVs are required to exchange data or services based on nested protocols. Unfortunately, it is difficult to do so because the protocols vary from one manufacturer to another. Thus, two UAVs having different protocols will not be able to communicate. It becomes even more constraining when it is

a collaborative mission in which UAVs operate synchronously or asynchronously. This challenge opens the following research tracks:

- **The development of a protocol converter** that allows us to convert a given FANET protocol into a desired protocol in order to ensure interoperability between non-homogeneous UAVs.
- **The standardization of UAV communication rules** in order to evolve towards a reference model used by manufacturers as well as end users.

### 10.3. Collective Navigation

Collective navigation is a commonly used practice for swarm mobility. It consists of flying UAVs in a tight formation to maintain network connectivity. Difficulties arise with a large and highly mobile swarm, where the positions and directions of the UAVs change relative to each other. The most frequent issues to address are collision and crash avoidance, obstacle detection, and the connectivity of the swarm [24]. A better way to control and harmonize the movements of the swarm is to use the "divide-and-conquer" technique [56]. It consists of dividing the swarm into several sub-swarms. Each sub-swarm is assigned a static formation with a well-defined polygon shape (regular, irregular, concave, convex, or complex). The UAVs inside the sub-swarms move and maintain a distance from each other without leaving the boundary of their formation, while the sub-swarms move in relation to their delimitation while remaining two by two disjoint, as shown in Figure 20.

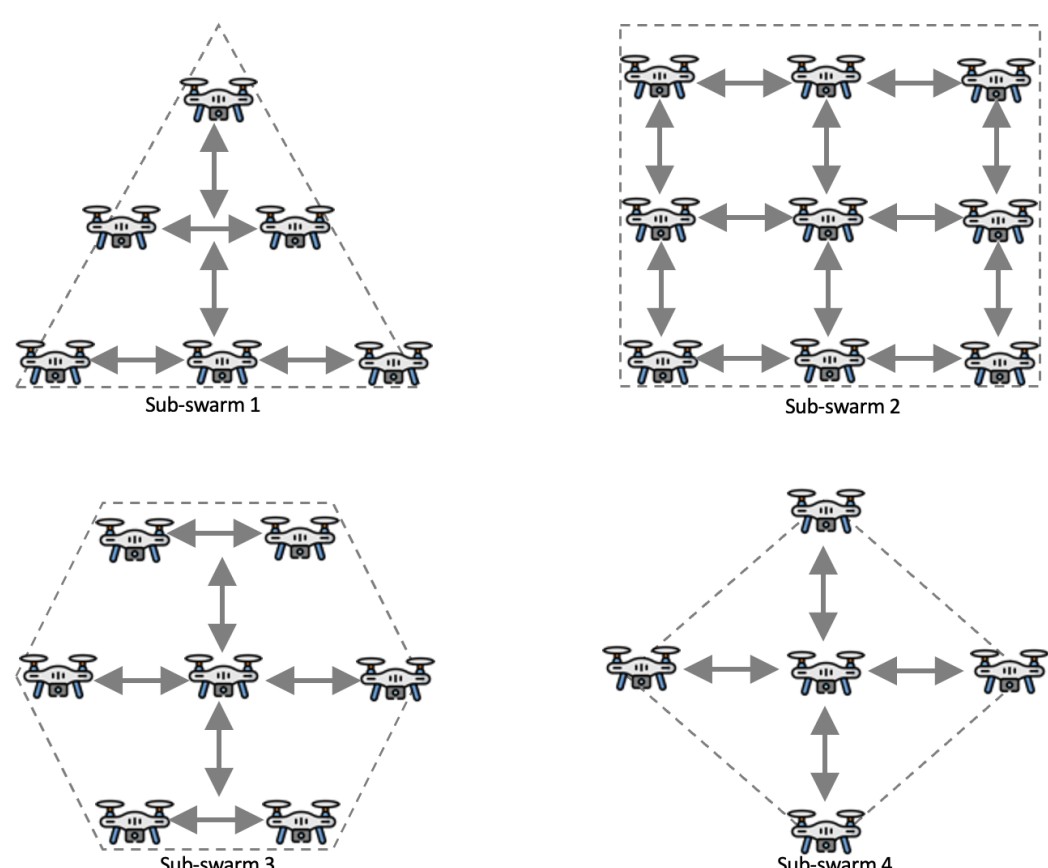

**Figure 20.** Example of swarm navigation based on divide-and-conquer.

### 10.4. Energy Consumption

To face the challenges of the maneuverability and portability of the UAVs, hardware components are reduced to a miniaturized size. As a result, they cannot carry many batteries or large-capacity batteries [41]. This constraint limits the energy consumption and contributes to the degradation of the communication and mobility performance of

UAVs [36]. The energy consumption related to the communication system is used to ensure the signal processing, the packet transmission, the routing protocol, and the data transmission of the onboard sensors. On the other hand, the power consumption of the propulsion system is much higher than the energy consumption for communication. It generates thrust and lift forces to ensure the mobility and movement of the UAVs in the air [41]. This depends on the type of UAV and number of rotors (see Section 3), payload weight, flying altitude, speed, and acceleration. In addition, there are environmental factors such as gravity, air density, wind conditions, and weather that affect the drone's energy consumption.

In the case of a multi-UAV system, the problem becomes even more challenging depending on the communication architecture, the mobility of the swarm, and the complexity of the mission. Several challenges remain to be addressed, including the optimization of the path planning [12,49,57], task optimization, and fast battery charging [40,42].

- **The optimization of the coverage paths** in order to reduce considerably the power consumption of the UAVs. This problem consists in finding the optimal paths that maximize the number of covered waypoints in a network while minimizing both energy costs and emissions [12,49]. A concrete application of this problem is the drone delivery operation, which consists in serving all the delivery waypoints without passing twice by the same point [38]. It is a multi-criteria optimization problem [58] that consists in determining the best energy consumption model that optimizes both the path cost, the energy consumed, and the emissions associated with the delivery strategy.

- **The efficient planning of the collaborative tasks** of the swarm. In a multi-UAV system, the tasks are running concurrently and in a parallel way to achieve a common goal. The UAVs share a large amount of information synchronously or asynchronously in the network for waypoint generation, map reconstruction, and collision avoidance [24,40]. These tasks are memory-intensive and also consume a significant amount of energy resources related to the communication system [36].

- **The maintenance of the battery** is another problem that affects the flight times of UAVs as the batteries used have a very limited lifespan [35,41]. It allows us to improve the battery life cycle and to extend its duration of use. Furthermore, this is one of the effective alternatives to have a battery considered safer and with a longer life cycle [35]. However, this task is complex because there is no exact method to determine the life of a battery and each type of UAV has its own maintenance pattern [42]. There are numerous factors affecting the life of the battery that must be considered, such as the amperage, voltage, temperature, type of charger, and charge/discharge cycles [37].

These challenges open future directions as follows:

- **The design of energy-efficient trajectory-based reinforcement learning**, which allows us to generate an adaptive path. Each UAV acts as an autonomous agent that readjusts its trajectory, speed, acceleration, and altitude to maximize energy gains. Moreover, it reinforces its adaptive capacity by using a multi-objective function that rewards the actions that minimize the cost of its path, its energy consumption, and its emissions. Some recent examples were presented in [36,40].

- **The optimization of the UAV flight tour inspired by combinatorial optimization and the integer programming problem**, in which, starting from the launch position of the UAV, we try to find an optimal set of paths that the swarm must traverse in order to return to its initial position. This is a problem of NP-Hard complexity [12,49,59] because we cannot pass twice through the same waypoint. Moreover, this problem becomes even more complex when we add the constraints of energy consumption and emissions. Typical examples were presented by Dorling et al. [39] and Febria et al. [12] for drone delivery services.

- **A novel energy harvester for powering UAVs** that will allow the possible recharge of an ultra-fast rechargeable battery onboard a UAV and an improvement in the lifespan. A research work conducted in this direction has been proposed in [35] and the analytical results show possible flight time improvements in the range of 3 to 16 min,

depending on the flight conditions. In terms of energy usage, it appears that decreasing the UAV size and payload weight may be more important than communication methods for enhanced energy efficiency.

### 10.5. Regulation

The legislation on the use of UAVs is covered by rules and procedures. Before launching a UAV, some important points about the type of UAV, the operating license, the area of exploration, and the mission must be checked. For details of the commission implementing European Union regulation 2019/947 of 24 May 2019 on the rules and procedures governing the operation of uncrewed aerial vehicles, see Table 5. UAV operating licenses are categorized into open, special, and requiring approval. Depending on the mission, an assessment of the operational requirements, risks, and preservation of confidentiality of information must be approved. In the case of swarms, the first requirement is to comply with the maximum number of UAVs allowed to fly simultaneously. Since a swarm can be composed of various types of UAVs, it is important to ensure compliance with the operational requirements and risks (collisions and signal interference) associated with handling these types of aerial vehicles. These non-technical challenges may provide an opportunity for research into legislation on the use of homogeneous and non-homogeneous swarms.

**Table 5.** Most important regulations on the use of UAVs for setting up a swarm.

| Category | Article | Regulation |
| --- | --- | --- |
| UAV categories | Article 3 | Uncrewed aircraft systems operational categories |
| | Article 4 | The UAS operational category "open" |
| | Article 5 | The UAS operational category "special" |
| | Article 6 | The UAS operating category "requiring approval" |
| Operational requirements | Article 7 | Rules and procedures for the operation of UAS |
| | Article 8 | Rules and procedures related to the competency of long-distance pilots |
| | Article 10 | Rules and procedures related to the airworthiness of UAS |
| Risk classes | Article 11 | Rules for operational risk assessment |
| | Article 12 | Operating license for the "special" category |
| | Article 14 | Registration of the UAS operator and UAS subject to licensing |
| Long distance flight | Article 13 | Cross-border operation or operation outside of the state of registration |
| | Article 15 | Operating conditions for UAS geographic areas |
| Privacy policy | Article 19 | Security information |

European Union regulation 2019/947 of 24 May 2019 (https://eur-lex.europa.eu/legal-content/EN/AUTO/?uri=CELEX:02019R0947-20220404, accessed on 15 March 2023).

## 11. Conclusions

Multi-UAV systems are of great interest to the scientific community because they represent a wide range of opportunities. Their usefulness in the industrial sector is no longer in doubt. They are an effective means for carrying out complex missions that require several types of UAVs. The question of the implementation of a multi-UAV system is a subject that involves two major challenges: technical–functional and legal/ethical. At the technical–functional level, the multi-UAV system must ensure interoperability between these components and maintain a level of autonomy that allows it to make collective decisions without human intervention. Using multi-UAVs for highly mobile network and wide-area missions is a challenging task because the UAVs must not move too far away from each other to avoid signal loss, which is not often the case. In addition, there are constraints related to the system infrastructure, the communication architecture, and the routing protocol. From a legal/ethical point of view, it is very important to consider people's privacy and safety, as well as to implement rigorously the regulations that guarantee the responsible use of multi-UAV systems.

Many scientific contributions have already explored countermeasures to these concerns; nevertheless, certain areas, such as mathematical modeling, the comparative study of communication architectures as well as routing protocols, and decision-making roadmap concerns, have yet to be addressed. With the growing interest in self-organizing swarms, these four challenges cannot be ignored. For this reason, this review paper offers a thorough investigation of the challenges associated with deploying an autonomous multi-UAV system that integrates collaborative actions, as well as the complexity of the collaborative target detection problem in distributed environments. It is an NP-Hard problem [58], which results in challenges related to the balancing of workloads in the swarm and the minimization of the path costs to reach targets. The current challenges, as well as a number of suggestions, are presented in this study, which emphasizes that the performance of multi-UAV systems can be improved if collaborative decision making, collective navigation, energy consumption, and regulation problems that remain to be addressed in this area are dealt with seriously.

**Author Contributions:** Conceptualization, W.Y.H.A. and S.L.; methodology, W.Y.H.A.; software, J.S.F.; validation, S.L., R.G. and M.B.; formal analysis, W.Y.H.A.; investigation, W.Y.H.A.; resources, W.Y.H.A. and J.S.F.; writing—original draft preparation, W.Y.H.A.; writing—review and editing, S.L.; visualization, R.G. and M.B.; supervision, S.L.; project administration, R.G. and M.B.; funding acquisition, R.G. and M.B. All authors have read and agreed to the published version of the manuscript.

**Funding:** This research was funded by CASUS Open X Projects.

**Data Availability Statement:** Not applicable.

**Conflicts of Interest:** The authors declare no conflicts of interest.

## Abbreviations

The following abbreviations are used in this manuscript:

| | |
|---|---|
| UAV | Uncrewed Aerial Vehicle |
| UAS | Uncrewed Aircraft System |
| RPA | Remotely Piloted Aircraft |
| RPAS | Remotely Piloted Aircraft System |
| M-UAV | Multi Uncrewed Aerial Vehicle |
| VTOL | Vertical Takeoff and Landing |
| IoT | Internet of Things |
| AI | Artificial Intelligence |
| MANET | Mobile Ad-Hoc Network |
| VANET | Vehicular Ad-Hoc Network |
| FANET | Flying Ad-Hoc Network |
| RC | Remote Controller |
| GCS | Ground Control Station |
| A2A | Air to Air |
| G2G | Group to Group |
| A2G | Air to Ground |
| OLSC | Oriented Line Segment Coverage |
| LCAD | Load Carry And Deliver |
| DCR | Data-Centric Routing |
| MLHR | Multi-Level Hierarchical Routing |
| OLSR | Optimized Link State Routing |
| DSDV | Destination Sequence Distance Vector |
| BATMAN | Better Approach to Mobile Ad-Hoc Network |
| DSR | Dynamic Source Routing |
| AODV | Ad-Hod On-Demand Vector |
| TSOD | Time-Slotted On-Demand |

|      |                                               |
|------|-----------------------------------------------|
| ZRP  | Zone Routing Protocol                         |
| TORA | Temporarily Ordered Routing Algorithm         |
| GPSR | Greedy Perimeter Stateless Routing            |
| GLSR | Geographic Load Share Routing                 |
| GPMOR| Geographic Position Mobility-Oriented Routing |
| RGR  | Reactive Greedy Reactive                      |
| CAN  | Clustering Algorithm for Networking           |
| MPC  | Mobility Prediction Clustering                 |
| IMU  | Inertial Measurement Unit                     |

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
