# Peer review of "Investigation of Autonomous Multi-UAV Systems for Target Detection in Distributed Environment: Current Developments and Open Challenges"

_drones, doi:10.3390/drones7040263_

Round 1

Reviewer 1 Report

Dear Authors,

please refer to the attached document. Thanks!

Author Response

We appreciate the comments from the reviewers. Thank you for reviewing our manuscript.

Response to Reviewer #1. We have incorporated all your suggestions into the revision. They were very helpful. Thank you.

Comment 1. I recommend to the authors to carry out the following changes:

(-) Although in general, the aim of the paper is clear, the abstract should be elegant and well written. It is opinion of this reviewer that the abstract requires a restructuring to have a better fluency. It should be better organized. This will assist the readers quickly to understand the pros and cons of this model. In fact, the part “In this manuscript, we propose a formal definition of the ecosystem of a multi-UAV system as well as a comparative study of UAV types and their operation mode. We also propose a mathematical modeling of the problem of collaborative drones in a distributed environment. We survey research works that highlight the use cases of multi-UAV systems. Then, we establish a comparative study of communication architectures and routing protocols in a UAV network. We provide multi-criteria decision-making roadmaps allowing to choose the architectures and the routing protocols adapted for specific missions. Finally, we discuss current challenges and future research opportunities in this area” should be more clear, better written, more harmonious and non-repetitive.

Response to Comment 1.  We have revised the abstract to make it more elegant and fluid in the understanding of the manuscript. The following changes in the abstract have been carried out according to your suggestions. 

Comment 2. Except for a part of abstract, the text is reasonably clear, so this reviewer recommends a minimum control of English language.

Response to Comment 2.  We have checked the new manuscript carefully. All spelling, grammar and typing errors have been corrected.

Comment 3. The authors need to mention and highlight what the current paper adds to the discussion, and in what way it is different from what was published before.

Response to Comment 3.  As suggested, we revised the presentation logic of the section 1 (Introduction). And for better understanding of the added values of our manuscript, we have added a subsection (lines 153-173) that addresses the gaps of works conducted about multi-UAV system integrated in collaborative actions.

Comment 4. In your interest, in the case of review manuscript please, you should put much more figures, in each section of the review. They are need in order to make the review more interesting for the reader.

Response to Comment 4.  The structure and figures of the revised manuscript were adapted as suggested by the submission guidelines of the journal. We have added a new figure (Figure 8 line 647) that provides a quick overview of the different multi-UAV communication architectures discussed in this paper.

Comment 5. I recommend to the authors to replace the old references with the recent papers published within the last five years to make their review paper up to date with the new trends in recent related research.

Response to Comment 5.  As recommended, we have included 16 relevant and recent references (including reviewers' proposals) to address the questions discussed in revised version of the manuscript:

Abro, Ghulam E. Mustafa, Saiful Azrin B. M. Zulkifli, Rana Javed Masood, Vijanth Sagayan Asirvadam, and Anis Laouti. “Comprehensive Review of UAV Detection, Security, and Communication Advancements to Prevent Threats.” Drones 6, no. 10 (October 2022): 284. https://doi.org/10.3390/drones6100284.

Chen, Yukai, Donkyu Baek, Alberto Bocca, Alberto Macii, Enrico Macii, and Massimo Poncino. “A Case for a Battery-Aware Model of Drone Energy Consumption.” In 2018 IEEE International Telecommunications Energy Conference (INâ„¡EC), 1–8, 2018. https://doi.org/10.1109/INTLEC.2018.8612333.

Chethan, Ragala, and Indrani Kar. “Multi-Agent Coverage Path Planning Using a Swarm of Unmanned Aerial Vehicles.” In 2022 IEEE 19th India Council International Conference (INDICON), 1–6, 2022. https://doi.org/10.1109/INDICON56171.2022.10039853.

Citroni, Rocco, Franco Di Paolo, and Patrizia Livreri. “A Novel Energy Harvester for Powering Small UAVs: Performance Analysis, Model Validation and Flight Results.” Sensors 19, no. 8 (January 2019): 1771. https://doi.org/10.3390/s19081771.

Dorling, Kevin, Jordan Heinrichs, Geoffrey G. Messier, and Sebastian Magierowski. “Vehicle Routing Problems for Drone Delivery.” IEEE Transactions on Systems, Man, and Cybernetics: Systems 47, no. 1 (January 2017): 70–85. https://doi.org/10.1109/TSMC.2016.2582745.

Galkin, Boris, Jacek Kibilda, and Luiz A. DaSilva. “UAVs as Mobile Infrastructure: Addressing Battery Lifetime.” IEEE Communications Magazine 57, no. 6 (June 2019): 132–37. https://doi.org/10.1109/MCOM.2019.1800545.

Hajijamali Arani, Atefeh, M. Mahdi Azari, Peng Hu, Yeying Zhu, Halim Yanikomeroglu, and Safieddin Safavi-Naeini. “Reinforcement Learning for Energy-Efficient Trajectory Design of UAVs.” IEEE Internet of Things Journal 9, no. 11 (June 2022): 9060–70. https://doi.org/10.1109/JIOT.2021.3118322.

Mehendale, Ninad. “Investigating the Battery Life Issues in Unmanned Aerial Vehicles: An Analysis of Challenges and Proposed Solutions.” SSRN Scholarly Paper. Rochester, NY, 2021. https://doi.org/10.2139/ssrn.4324196.

Zhang, Juan, James F. Campbell, Donald C. Sweeney II, and Andrea C. Hupman. “Energy Consumption Models for Delivery Drones: A Comparison and Assessment.” Transportation Research Part D: Transport and Environment 90 (January 1, 2021): 102668. https://doi.org/10.1016/j.trd.2020.102668.

Zhang, Shilei, and Taining Cheng. “Optimizing Energy Consumption of Rotor UAV by Path Planning.” In 2022 Global Conference on Robotics, Artificial Intelligence and Information Technology (GCRAIT), 54–58, 2022. https://doi.org/10.1109/GCRAIT55928.2022.00020.

Choudhary, Gaurav, Vishal Sharma, Takshi Gupta, Jiyoon Kim, and Ilsun You. “Internet of Drones (IoD): Threats, Vulnerability, and Security Perspectives.” Research Briefs on Information and Communication Technology Evolution 4 (August 15, 2018): 64–77.

Digulescu, Angela, Cristina Despina-Stoian, Denis Stănescu, Florin Popescu, Florin Enache, Cornel Ioana, Emanuel Rădoi, Iulian Rîncu, and Alexandru Șerbănescu. “New Approach of UAV Movement Detection and Characterization Using Advanced Signal Processing Methods Based on UWB Sensing.” Sensors 20, no. 20 (January 2020): 5904. https://doi.org/10.3390/s20205904.

Febria, Jessi, Christine Dewi, and Evangs Mailoa. “Comparison of Capacitated Vehicle Routing Problem Using Initial Route and Without Initial Route for Pharmaceuticals Distribution.” In 2021 2nd International Conference on Innovative and Creative Information Technology (ICITech), 94–98, 2021. https://doi.org/10.1109/ICITech50181.2021.9590116.

Liu, Yao, Zhihao Luo, Zhong Liu, Jianmai Shi, and Guangquan Cheng. “Cooperative Routing Problem for Ground Vehicle and Unmanned Aerial Vehicle: The Application on Intelligence, Surveillance, and Reconnaissance Missions.” IEEE Access 7 (2019): 63504–18. https://doi.org/10.1109/ACCESS.2019.2914352.

Maini, Parikshit, Kaarthik Sundar, Mandeep Singh, Sivakumar Rathinam, and P. B. Sujit. “Cooperative Aerial–Ground Vehicle Route Planning With Fuel Constraints for Coverage Applications.” IEEE Transactions on Aerospace and Electronic Systems 55, no. 6 (December 2019): 3016–28. https://doi.org/10.1109/TAES.2019.2917578.

Shakhatreh, Hazim, Ahmad H. Sawalmeh, Ala Al-Fuqaha, Zuochao Dou, Eyad Almaita, Issa Khalil, Noor Shamsiah Othman, Abdallah Khreishah, and Mohsen Guizani. “Unmanned Aerial Vehicles (UAVs): A Survey on Civil Applications and Key Research Challenges.” IEEE Access 7 (2019): 48572–634. https://doi.org/10.1109/ACCESS.2019.2909530.

Comment 6.  The number of recent articles is very limited in case the article read is a review. For this reason, only to add more information, this reviewer suggests some papers but more papers published within the maximum five years is needed for one review

- Abro, G.E.M.; Zulkifli, S.A.B.M.; Masood, R.J.; Asirvadam, V.S.; Laouti, A. Comprehensive Review of UAV Detection, Security, and Communication Advancements to Prevent Threats. Drones 2022, 6, 284.

Referring to the part “Energy consumption” page 3 of 35,

An energy-harvesting model is proposed to power supply small UAVs, known as micro air vehicles (MAVs) where analytical results show possible flight time improvements in the range of 3 to 16 min, depending on the flight conditions. Concerning energy consumption, reducing UAV and payload weight seems to be more significant with communication protocols for improved energy efficiency.

- Citroni, R.; Di Paolo, F.; Livreri, P. A Novel Energy Harvester for Powering Small UAVs: Performance Analysis, Model Validation and Flight Results. Sensors 2019, 19, 1771.

Response to Comment 6.  As suggested, we have included the two references to better justify the challenges related to energy consumption and security and collision avoidance issues. These references were added in the new subsection (10.4 Energy consumption).

Comment 7.  The keywords are appropriate. However, this is only a reviewer's opinion. There is a limited numbers of key words for this type of journal.

Response to Comment 6.  We added 4 keywords (UAS; Autonomous Aerial Vehicles; Collaborative Missions; Distributed Environment; Distributed Path Planning) that better fit the scope of the article and the journal.

Comment 7.  Although there is a special section at the end of the paper used to insert all the full forms and their abbreviations, in the text several acronyms are without corresponding meaning. Please, check the whole paper.

Response to Comment 7.   We have corrected all inconsistencies in the list of abbreviations.

Comment 8.  In the text, figure 10 is present, as caption but it is not indicated in the text.

Response to Comment 8.   In the new version of the manuscript, the figure has been indicated in section 6.2 (line 664), thank you very much.

                                        Figure 10 => Figure 11

Comment 9.  Finally, this question represents a curiosity of this reviewer. Why do the authors instead of using the common term "Unmanned" use "Uncrewed" in the whole paper? Thanks!

Response to Comment 9.   Recently the drone community has recommended to use the term "Uncrewed" instead of "Unmanned" as it is considered gender neutral. The term "Uncrewed"  is a new non-gender-based terminology. More information is provided is the following link : https://www.canada.ca/en/department-national-defence/maple-leaf/defence/2021/05/unmanned-to-uncrewed-moving-away-from-gender-based-terminology.html

Reviewer 2 Report

Could you provide some simulation results?

One of the most critical issues in UAV networks is energy consumption. Please provide more information about it. The following paper is a good reference for it:

"Reinforcement Learning for Energy-Efficient Trajectory Design of UAVs," IEEE Internet of Things Journal, 2022.

Author Response

We thank the reviewer for the positive feedback. We agree that this paper will be of interest to the readership of the journal.

Response to Reviewer #2. We appreciate the useful comments from the reviewer.
Comment 1. Could you provide some simulation results?

Response to Comment 1.  This one will be the subject of another work because as we showed it, there are 4 architectures of communication with various protocols (17 in the case of this paper). Each protocol operates according to a paradigm that is specific to the communication architecture. We have planned a paper dedicated to this task.

Comment 2. One of the most critical issues in UAV networks is energy consumption. Please provide more information about it. The following paper is a good reference for it:  "Reinforcement Learning for Energy-Efficient Trajectory Design of UAVs," IEEE Internet of Things Journal, 2022.

Response to Comment 2.  We cited the reference in the introduction (line 126) and section 10.4 "Energy consumption" (lines 1021, 1048 and 1063)

Reviewer 3 Report

This manuscript presents a discussion on the challenges and future trends of multi-UAV system integrated collaborative actions. So, this manuscript is a review article rather than a research article. Therefore, the authors should revise this draft according to the review research article. However, if the authors want to write it as a research article, they must present the proposed method and experimental results and compare this method to others.

Besides, I want to give some comments to improve this version as follows:

The authors must present the innovations and contributions of this research in the introduction section.

The authors should review several related works regarding autonomous MUAV systems for target detection to determine the research gap and how to bridge it.

The author should shorten part 2; the author wrote this part too long and unimportant.

There are several methods to classify UAVs. So, the authors should present them. The author should show how and what types of UAVs are classified within each method. 

The authors must revise the conclusion section to present the key results of this research.

Author Response

We appreciate the positive feedback. Thank you!!!

Response to Reviewer #3: We appreciate the comments from the reviewers. Thank you for reviewing our manuscript.

Comment 1. This manuscript presents a discussion on the challenges and future trends of multi-UAV system integrated collaborative actions. So, this manuscript is a review article rather than a research article. Therefore, the authors should revise this draft according to the review research article. However, if the authors want to write it as a research article, they must present the proposed method and experimental results and compare this method to others.

Response to Comment 1.  This manuscript is a survey paper.

Comment 2. The authors must present the innovations and contributions of this research in the introduction section.

Response to Comment 2.  As recommended, we updated the introduction presentation logic. Also, a part (lines 153-173) that fills in the gaps of conducted work on the use of multi-UAV systems in cooperative missions has been included to help readers better grasp the added value and contribution of this study.

Comment 3. The authors should review several related works regarding autonomous MUAV systems for target detection to determine the research gap and how to bridge it.

Response to Comment 3.  We discussed these aspects in section 5. Unfortunately, there is little research that addresses this topic. To answer this question, we have discussed the gaps, challenges, and future directions regarding autonomous multi-UAV. These discussions focused on collaborative decision making, communication infrastructure, collective navigation, energy consumption  and regulation.

Comment 4. The author should shorten part 2; the author wrote this part too long and unimportant.

Response to Comment 4.  Part 2 has been divided into two sections: Section 2 (Background ) and Section 3 (Classification of UAVs). Thank you very much!!!

Comment 5. There are several methods to classify UAVs. So, the authors should present them. The author should show how and what types of UAVs are classified within each method.

Response to Comment 5.  This part is a multi-criteria study and will be the subject of a separate work followed by a publication with results. There is no one standard when it comes to the classification of UAV.  Defense agencies have their own standard, and civilians have their ever-evolving loose categories . People classify them according to design configuration (1) such as size, weight, engine type or power system; or according to the degree of operational autonomy (2) such as payload capability, range, endurance, maximum flight altitude, operational role, etc. In this work, we have opted for a classification based on size because it is the most used and encompasses several measurement characteristics.

Comment 6. The authors must revise the conclusion section to present the key results of this research.

Response to Comment 6.  The conclusion has been modified according to the above remarks. We have highlighted the main contributions of this paper and the remaining challenges for the effective overuse of multi-UAV systems.

Round 2

Reviewer 1 Report

Dear Authors,

please refer to the attached document. Thanks!

Reviewer 3 Report

The author has made some improvements to the original manuscript. I think it could be published after some polish and refinement.